# Open-Text Aerial Detection: A Unified Framework for Aerial Visual Grounding and Detection

Guoting Wei [1 4 *]   Xia Yuan [1 *]   Yang Zhou [2]   Haizhao Jing [2]   Yu Liu [3]
Xianbiao Qi [4]   Chunxia Zhao [1]   Haokui Zhang [2 4 †]   Rong Xiao [4]

## Abstract

Open-Vocabulary Aerial Detection (OVAD) and Remote Sensing Visual Grounding (RSVG) have emerged as two key paradigms for aerial scene understanding. However, each paradigm suffers from inherent limitations when operating in isolation: OVAD is restricted to coarse category-level semantics, while RSVG is structurally limited to single-target localization. These limitations prevent existing methods from simultaneously supporting rich semantic understanding and multi-target detection. To address this, we propose OTA-Det, the first unified framework that bridges both paradigms into a cohesive architecture. Specifically, we introduce a task reformulation strategy that unifies task objectives and supervision mechanisms, enabling joint training across datasets from both paradigms with dense supervision signals. Furthermore, we propose a dense semantic alignment strategy that establishes explicit correspondence at multiple granularities, from holistic expressions to individual attributes, enabling fine-grained semantic understanding. To ensure real-time efficiency, OTA-Det builds upon the RT-DETR architecture, extending it from closed-set detection to open-text detection by introducing several highly efficient modules, achieving state-of-the-art performance on six benchmarks spanning both OVAD and RSVG tasks while maintaining real-time inference at 34 FPS. Code is available at https://github.com/GT-Wei/OTA-Det.

*Equal contribution  [1]Nanjing University of Science and Technology, Nanjing, China [2]Northwestern Polytechnical University, Xi'an, China [3]Zhejiang Lab, Hangzhou, China [4]Intellifusion, Shenzhen, China. Correspondence to: Haokui Zhang <hkzhang@nwpu.edu.cn>.

*Proceedings of the $43^{rd}$ International Conference on Machine Learning*, Seoul, South Korea. PMLR 306, 2026. Copyright 2026 by the author(s).

## 1. Introduction

Aerial object detection and visual grounding are essential for diverse applications ranging from UAV-based embodied AI to practical scenarios such as logistics management and traffic surveillance (Liu et al., 2023; Adão et al., 2017; San et al., 2018; Tao et al., 2025). Traditional closed-set aerial object detection (Cheng et al., 2023; Sun et al., 2022b; Xu et al., 2024; Shi et al., 2023; Cai et al., 2024) is increasingly inadequate for these scenarios due to rigid category constraints. In response, Open-Vocabulary Aerial Detection (OVAD) and Remote Sensing Visual Grounding (RSVG)[1] have emerged as two prominent paradigms that establish associations between visual features and textual semantics, better aligning with real-world application needs.

These paradigms are typically studied as separate tasks for different scenarios and exhibit inherent limitations when operating in isolation. As illustrated in Fig. 1, OVAD (Huang et al., 2025c; Pan et al., 2025; Li et al., 2024b; Wei et al., 2024) focuses on detecting all instances matching arbitrary category-level text inputs but is limited to coarse category-level semantics, lacking the capacity to interpret sophisticated linguistic descriptions. In contrast, RSVG (Zhan et al., 2023; Li et al., 2024a; Liu et al., 2025) excels at precisely localizing targets by comprehending complex referring expressions, yet is structurally restricted to single-target localization due to its Referring Expression Comprehension (REC) formulation (Mao et al., 2016).

These limitations create a gap between their capabilities and practical requirements. In real-world applications, these two scenarios often coexist. Users naturally provide textual inputs with multi-granular semantics, ranging from simple category names (e.g., "vehicle") to complex referring expressions (e.g., "the red truck parked near the warehouse entrance"), and expect to detect all targets matching the given descriptions. However, existing paradigms fail to satisfy these requirements when applied individually. An intuitive solution is to unify both paradigms, integrating their

---

[1]In this paper, RSVG refers to visual grounding in both drone and remote sensing images, as they share similar task formulations and challenges.

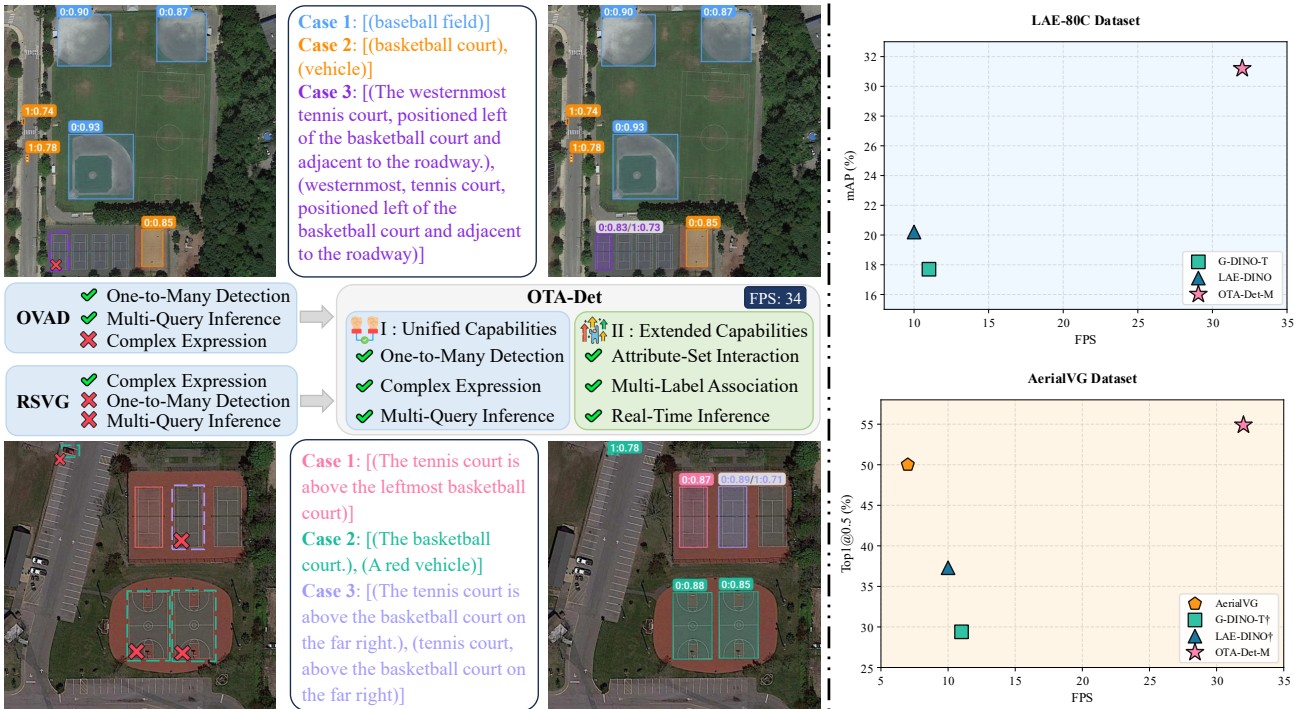

*Figure 1.* Comparison of task paradigms and capabilities. **Left:** Top two images show comparison between OVAD and OTA-Det detection results. Bottom two images show comparison between RSVG and OTA-Det detection results. OVAD supports One-to-Many Detection and Multi-Query Inference but is limited to category-level semantics, while RSVG handles Complex Expressions but is restricted to Single-Target scenarios. OTA-Det combines the strengths of both paradigms with extended capabilities including Attribute-Set Interaction, Multi-Label Association, and Real-Time Inference at 34 FPS. **Right:** Performance comparison on LAE-80C (top) and AerialVG (bottom) benchmarks, demonstrating OTA-Det's superior accuracy and efficiency.

complementary strengths to enable comprehensive multi-granular semantic understanding and multi-target detection.

However, realizing this unification is non-trivial and faces two fundamental challenges at different levels. ***(i) At the task formulation level,*** OVAD and RSVG exhibit fundamental inconsistency in task objectives. OVAD performs joint classification and localization for all instances matching given category names, while RSVG focuses exclusively on localizing single targets corresponding to individual referring expressions. ***(ii) At the semantic alignment level,*** OVAD establishes explicit vision-language correspondence through contrastive learning, whereas RSVG relies on implicit multimodal interaction with localization-only supervision. More critically, existing RSVG methods align objects exclusively with holistic sentence representations, neglecting fine-grained correspondence with individual attributes. This risks semantic pseudo-alignment where models bind visual features to complete sentences without understanding constituent attributes. For instance, given "a green taxi on the left," models may successfully localize the target while failing to capture individual semantics of category, color, and location, thereby limiting compositional generalization and fine-grained semantic mining from limited data.

To address these challenges, we propose Open-Text Aerial Detection (OTA-Det), a unified framework that bridges OVAD and RSVG within a cohesive architecture. As shown in Fig. 1, OTA-Det not only inherits the complementary strengths of both paradigms but also introduces several extended capabilities. Our contributions are as follows:

- **Task Reformulation Strategy.** To address the task formulation inconsistency, we reformulate RSVG from pure localization into a joint classification-localization problem, structurally aligning it with OVAD while preserving expression comprehension capability. Furthermore, to enable unified semantic alignment, we propose an Image-Level Annotation Aggregation strategy that provides dense classification supervision, enabling explicit vision-language correspondence through contrastive learning.

- **Dense Semantic Alignment Strategy.** To mitigate semantic pseudo-alignment, we introduce a dense semantic alignment mechanism comprising Attribute-Level Data Decomposition, a Unified Correspondence Matrix, and a Decoupled Multi-Granular Head. This strategy enables explicit alignment at multiple granularities, from holistic sentence representations to indi-

vidual attribute components, ensuring comprehensive visual-semantic correspondence and enabling attribute-set interaction for flexible compositional queries.

- **Real-Time Unified Architecture.** We propose OTA-Det, a unified framework that builds upon the RT-DETR architecture, extending it from closed-set detection to open-text detection. By leveraging offline text encoding and introducing several highly efficient modules, OTA-Det achieves real-time inference at 34 FPS while delivering state-of-the-art performance on six benchmarks spanning both OVAD and RSVG tasks.

**Conflict of Interest Disclosure.** All datasets and benchmarks used in this paper are publicly available, and the authors declare no competing interests.

## 2. Preliminary

To address the challenges of unifying OVAD and RSVG, we first formally review the formulations of existing aerial detection paradigms, clarifying their fundamental discrepancies in task objectives and semantic alignment mechanisms.

**Closed-Set Aerial Detection** processes a single aerial image $I \in \mathbb{R}^{H \times W \times 3}$ and predicts all objects belonging to a predefined category set $\mathcal{C}_{fixed}$. The task is formulated as:

$$I \to \mathcal{D} = \{(b_i, c_i, s_i)\}_{i=1}^{N}, \quad c_i \in \mathcal{C}_{fixed}, \qquad (1)$$

where $b_i \in \mathbb{R}^4$ denotes bounding box coordinates, $c_i$ is the predicted class label, and $s_i \in [0, 1]$ represents confidence scores. The predictions are supervised by ground truth $\mathcal{G} = \{(b_j^*, c_j^*)\}_{j=1}^{M}$ via classification and localization losses.

**Open-Vocabulary Aerial Detection (OVAD)** breaks the category limitation through explicit visual-semantic alignment via contrastive learning, thereby enabling it to handle arbitrary category names. Given an image $I$ and a set of arbitrary categories $\mathcal{T}_{category} = \{t_1, t_2, \ldots, t_k\}$ where each $t_i$ can be a novel class unseen during training, the task is defined as:

$$(I, \mathcal{T}_{category}) \to \mathcal{D} = \{(b_i, c_i, s_i)\}_{i=1}^{N}, \quad c_i \in \mathcal{T}_{category}, \tag{2}$$

where $s_i \in [0, 1]$ represents the visual-semantic similarity score. The predictions are supervised by ground truth $\mathcal{G} = \{(b_j^*, t_j^*)\}_{j=1}^{M}$ via classification and localization losses.

**Remote Sensing Visual Grounding (RSVG)** implicitly bridges the visual-semantic relationship through cross-modal interaction with localization-only supervision. It specializes in understanding complex referring expressions $E$ and localizing the corresponding single target within an image $I$. The task formulation is:

$$(I, E) \to \mathcal{D} = \{b\}, \quad b \in \mathbb{R}^4, \tag{3}$$

where $\mathcal{D}$ is a single-element set containing the bounding box $b$ of the unique target described by $E$. The prediction is supervised by ground truth $\mathcal{G} = \{b^*\}$ via localization loss.

## 3. Method

In this section, we first introduce RSVG Task Reformulation (Sec. 3.1) that aligns task objectives between the two paradigms. Then, we propose Dense Semantic Alignment (Sec. 3.2) to establish fine-grained visual-semantic correspondence. Finally, we present the complete OTA-Det architecture in Sec. 3.3.

### 3.1. Task Reformulation Strategy

Comparing Eq. 2 and Eq. 3, OVAD and RSVG exhibit two fundamental discrepancies: *(i) Task Objective Discrepancy.* OVAD performs joint classification-localization with outputs $\mathcal{D} = \{(b_i, c_i, s_i)\}$, while RSVG outputs only bounding boxes $b$. To align task objectives, we first propose Naive Reformulation to introduce classification supervision. *(ii) Supervision Density Discrepancy.* OVAD accepts a category set $\mathcal{T}_{category} = \{t_1, \ldots, t_k\}$ with dense supervision, whereas RSVG processes single expressions $E$, yielding sparse supervision that hinders discriminative learning. To achieve dense supervision comparable to OVAD, we further propose Image-Level Annotation Aggregation to restructure the data organization from sentence level to image level.

#### 3.1.1. Naive Reformulation

To introduce classification supervision, a straightforward approach is to reformulate RSVG as a joint classification-localization task. Specifically, we treat the single expression $E$ as a single-element set, *i.e.*, $\mathcal{T}_E = \{E\}$, with ground truth $\mathcal{G} = \{(b^*, E)\}$. The reformulated task becomes:

$$(I, \mathcal{T}_E) \to \mathcal{D} = \{(b_i, c_i, s_i)\}_{i=1}^{N}, \quad c_i \in \mathcal{T}_E, |\mathcal{T}_E| = 1. \tag{4}$$

This reformulation enables explicit visual-semantic alignment through contrastive learning, similar to OVAD. However, the constraint $|\mathcal{T}_E| = 1$ means each sample still provides only a single supervision signal, failing to address the supervision density discrepancy identified above.

#### 3.1.2. Image-Level Annotation Aggregation

To address the sparse supervision limitation of Naive Reformulation, we restructure the data organization from sentence level to image level, as illustrated in Fig. 2(A). Specifically, we aggregate all referring expressions associated with a single image $I$ into a unified query set $\mathcal{T}_E = \{E_1, E_2, \ldots, E_K\}$, along with their corresponding bounding boxes. The ground truth is reformulated as

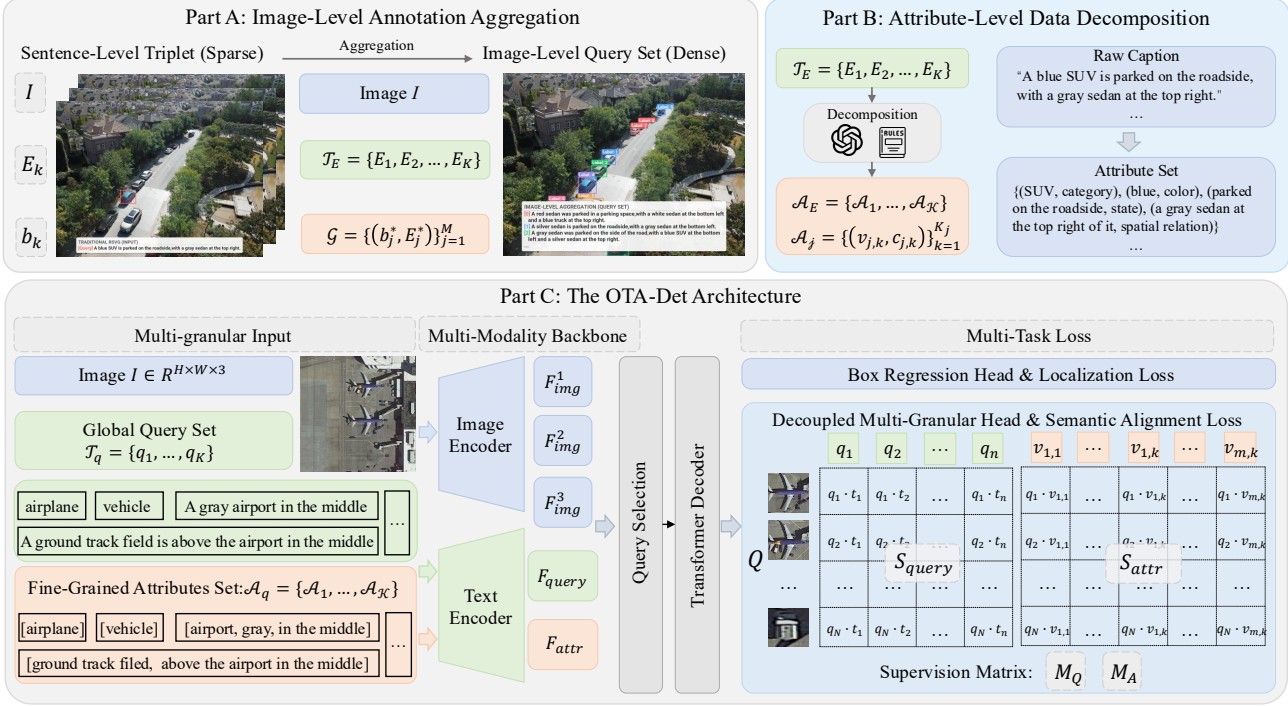

*Figure 2.* **Overview of the proposed OTA-Det framework. (A) Image-Level Annotation Aggregation** restructures sparse sentence-level triplets $\langle I, E_k, b_k \rangle$ into dense image-level query sets $\mathcal{T}_E$ with labeled ground truth $\mathcal{G}$, transforming RSVG into a joint classification-localization task structurally aligned with OVAD. **(B) Attribute-Level Data Decomposition** leverages an LLM to parse referring expressions in $\mathcal{T}_E$ into structured, target-centric attribute sets $\mathcal{A}_E$, enabling fine-grained alignment and mitigating semantic pseudo-alignment. **(C) The OTA-Det Architecture** processes multi-granular inputs through a Multi-Modality Backbone and employs a Decoupled Multi-Granular Head to compute independent similarity logits for holistic queries ($\mathbf{S}_{query}$) and fine-grained attributes ($\mathbf{S}_{attr}$). The Unified Correspondence Matrices $\mathbf{M}_Q$ and $\mathbf{M}_A$ serve as supervision targets, optimized via the MAL objective.

$\mathcal{G} = \{(b_j^*, E_j^*)\}_{j=1}^M$, and the task becomes:

$$(I, \mathcal{T}_E) \rightarrow \mathcal{D} = \{(b_i, c_i, s_i)\}_{i=1}^N, \quad c_i \in \mathcal{T}_E. \quad (5)$$

Comparing Eq. 5 with Eq. 2, this reformulation structurally aligns RSVG with OVAD, enabling explicit semantic classification through contrastive learning while supporting multi-query inference and one-to-many detection. Meanwhile, this unified formulation allows joint training on both OVAD and RSVG datasets to achieve multi-granularity semantic understanding spanning from category names to complex referring expressions.

**3.2. Dense Semantic Alignment Strategy**

After task reformulation, we enable explicit visual-semantic alignment by encoding each expression $E$ into a semantic token and applying contrastive learning. As discussed in Sec. 1, holistic alignment risks semantic pseudo-alignment where models bind visual features to complete sentences without understanding constituent attributes. To establish alignment at multiple semantic granularities, we propose a dense semantic alignment strategy that operates at both the data and architectural levels.

At the data level, we introduce: (i) Attribute-Level Data Decomposition extracts structured attributes from referring expressions, and (ii) Unified Correspondence Matrix encodes supervision signals across semantic granularities. At the architectural level, we design a Decoupled Multi-Granular Head (detailed in Sec. 3.3.3) to compute independent alignment scores for holistic queries and attributes.

### 3.2.1. ATTRIBUTE-LEVEL DATA DECOMPOSITION

To establish the data foundation for attribute-level alignment, we leverage an LLM to automatically decompose holistic expressions into structured, target-centric attribute sets.

**LLM Prompting Strategy.** The prompting strategy is governed by three key principles: (i) *Target-Centric Principle.* Following the REC task formulation, we extract attributes exclusively for the primary target, treating other entities as spatial references. (ii) *Verbatim Extraction.* To prevent hallucination and maintain semantic consistency, all extracted attributes must be exact substrings from the original caption without any rephrasing or paraphrasing. (iii) *Structured Categorization.* To ensure systematic attribute organization, we require the LLM to classify each attribute into specific

semantic types (e.g., category, color, spatial position).

**Attribute Decomposition.** For each expression $E_j \in \mathcal{T}_E$, the decomposition yields an attribute set $\mathcal{A}_j = \{(v_{j,k}, \tau_{j,k})\}_{k=1}^{K_j}$, where $v_{j,k}$ denotes the $k$-th attribute value text and $\tau_{j,k}$ denotes its semantic type. We construct a unified image-level attribute set $\mathcal{A}_E = \{\mathcal{A}_1, \mathcal{A}_2, \ldots, \mathcal{A}_K\}$ containing all attributes from expressions in $\mathcal{T}_E$.

### 3.2.2. UNIFIED CORRESPONDENCE MATRIX

Existing OVAD and RSVG follow a single-label classification scenario, where each target corresponds to only a single semantic entity (either a category name or a referring expression). However, with the introduction of multi-granularity semantics, each object now corresponds not only to a holistic query but also to multiple fine-grained attributes simultaneously, transforming the task into a multi-label classification scenario. To support this, we propose a Unified Correspondence Matrix that explicitly encodes all visual-semantic relationships. Specifically, we define two supervision matrices for each image:

**(i) Object-Query Matrix** $\mathbf{M}_Q \in \{0,1\}^{N_{obj} \times N_{query}}$ encodes binary correspondences between $N_{obj}$ ground-truth objects and $N_{query}$ holistic queries.

**(ii) Object-Attribute Matrix** $\mathbf{M}_A \in \{0,1\}^{N_{obj} \times N_{attr}}$ encodes correspondences between objects and $N_{attr}$ fine-grained attributes from the unified attribute set $\mathcal{A}_E$.

These matrices provide a unified representation of supervision signals across granularities and explicitly support two key capabilities through their geometric structure: *Row-wise*, the vector $\mathbf{M}_{i,:}$ encodes multi-label association, where $\sum_j \mathbf{M}_{ij} > 1$ indicates that object $i$ simultaneously matches multiple semantic entities. *Column-wise*, the vector $\mathbf{M}_{:,j}$ indicates one-to-many detection, where $\sum_i \mathbf{M}_{ij} > 1$ means query $j$ grounds to multiple object instances.

**Query-Attribute Hierarchical Mapping.** To preserve the hierarchical relationship between holistic queries and their decomposed attributes, we maintain an additional mapping matrix $\mathbf{M}_{map} \in \{0,1\}^{N_{query} \times N_{attr}}$, where $\mathbf{M}_{map}(i,j) = 1$ indicates that attribute $j$ belongs to query $i$. This matrix enables aggregating attribute-level predictions back to the query level during inference, ensuring the model captures the compositional structure of complex expressions.

### 3.3. OTA-Det Architecture

Building upon the reformulated task and multi-granular supervision, this section presents the OTA-Det architecture. We adopt DEIMv2 (Huang et al., 2025a), a variant of RT-DETR, as the base detector for its high efficiency and concise structure. As illustrated in Figure 2, the architecture accepts multi-granular inputs and introduces three key components: (i) a Multi-Modality Backbone designed to encode texts at multiple semantic granularities; (ii) a Decoupled Multi-Granular Head that enables simultaneous alignment of holistic query representations and fine-grained attribute representations; and (iii) a Multi-Task Loss that supervises these unified objectives.

### 3.3.1. UNIFIED TASK INTERFACE

To enable seamless processing of both OVAD and RSVG tasks within a single model, we formalize the unified input-output interface.

**Multi-granular Inputs.** The inputs consist of an aerial image $I \in \mathbb{R}^{H \times W \times 3}$, a query set $\mathcal{T}_q = \{q_1, \ldots, q_K\}$ where each $q_i$ can be either a category name or a referring expression, and a unified attribute set $\mathcal{A}_q$ containing all attributes corresponding to $\mathcal{T}_q$. Note that for category names, the category itself serves as its attribute.

**Training Outputs.** During training, the model generates dual-granularity similarity logits that align with the supervision matrices defined in Sec. 3.2:

$$(I, \mathcal{T}_q, \mathcal{A}_q) \rightarrow \{\mathbf{S}_{\text{query}}, \mathbf{S}_{\text{attr}}, \mathbf{B}\}, \qquad (6)$$

where $\mathbf{S}_{\text{query}} \in \mathbb{R}^{N_{pred} \times N_{query}}$ represents the query-level similarity logits supervised by the Object-Query Matrix $\mathbf{M}_Q$, $\mathbf{S}_{\text{attr}} \in \mathbb{R}^{N_{pred} \times N_{attr}}$ represents the attribute-level similarity logits supervised by the Object-Attribute Matrix $\mathbf{M}_A$, and $\mathbf{B} \in \mathbb{R}^{N_{pred} \times 4}$ denotes the predicted bounding boxes. Here $N_{pred}$ is the number of predicted objects.

**Inference Outputs.** During inference, the model produces multi-granular detection results:

$$(I, \mathcal{T}_q, \mathcal{A}_q) \rightarrow \mathcal{D} = \{(b_i, c_i, s_i, \mathbf{c}_i^{attr}, \mathbf{s}_i^{attr})\}_{i=1}^N, \quad (7)$$

where $c_i \in \mathcal{T}_q$ and $s_i \in [0,1]$ denote the query-level classification and similarity score, while $\mathbf{c}_i^{attr}$ and $\mathbf{s}_i^{attr}$ represent the attribute-level classifications and similarity scores, with $c_i^{(k)} \in \mathcal{A}_q$ and $s_i^{(k)} \in [0,1]$.

This dual-granularity output enables users to query targets either through holistic queries (via $c_i, s_i$) or through flexible attribute combinations (via $\mathbf{c}_i^{attr}, \mathbf{s}_i^{attr}$).

### 3.3.2. MULTI-MODALITY BACKBONE

Given the multi-granular inputs $(I, \mathcal{T}_q, \mathcal{A}_q)$, the backbone extracts visual features from the image and encodes textual features from both query and attribute sets:

$$\mathbf{F}_{\text{img}} = \Phi_{\text{img}}(I), \quad \mathbf{F}_{\text{query}} = \Phi_{\text{txt}}(\mathcal{T}_q), \quad \mathbf{F}_{\text{attr}} = \Phi_{\text{txt}}(\mathcal{A}_q). \qquad (8)$$

**Image Encoder.** We adopt DINOv3-STAs (Siméoni et al., 2025; Huang et al., 2025a) as the image encoder $\Phi_{\text{img}}$ to extract multi-scale features $\{\mathbf{F}_{\text{img}}^1, \mathbf{F}_{\text{img}}^2, \mathbf{F}_{\text{img}}^3\}$ at strides

$\{8, 16, 32\}$, which are subsequently fused through the hybrid encoder from RT-DETR (Zhao et al., 2024).

**Text Encoder.** For the text encoder $\Phi_{\text{txt}}$, we utilize the frozen SigLIPv2 (Tschannen et al., 2025) model to encode both holistic queries $\mathcal{T}_q$ and fine-grained attributes $\mathcal{A}_q$ into unified semantic representations.

### 3.3.3. DECOUPLED MULTI-GRANULAR HEAD

To replace the traditional classification head with semantic understanding, we design a contrastive head to compute vision-language semantic similarity via a function $\mathcal{S}(\cdot)$. Formally, given visual features $\mathbf{V}$ and text embeddings $\mathbf{T}$, the similarity logits are computed as:

$$\mathcal{S}(\mathbf{V}, \mathbf{T}) = \alpha \cdot \frac{\psi(\mathbf{V})\mathbf{T}^{\top}}{\|\psi(\mathbf{V})\|_2 \|\mathbf{T}\|_2} + \beta + \mathcal{M}, \qquad (9)$$

where $\psi(\cdot)$ is a linear projection layer mapping visual features to align with the text embedding space, $\alpha$ is a learnable logit scale factor, and $\beta$ is a learnable bias term. $\mathcal{M}$ denotes the padding mask used to suppress invalid text tokens introduced for parallel computation (set to $-\infty$).

To enhance interaction and understanding at the attribute level, we decouple the alignment process by computing similarity logits independently for holistic queries and fine-grained attributes. This design is motivated by the distinct semantic granularities between complete descriptions and specific attribute properties. Decoupled processing allows for more effective feature optimization while preventing mutual interference. Formally, given decoder query features $\mathbf{Q} \in \mathbb{R}^{N_q \times D}$, we compute:

$$\mathbf{S}_{\text{query}} = \mathcal{S}(\mathbf{Q}, \mathbf{F}_{\text{query}}), \quad \mathbf{S}_{\text{attr}} = \mathcal{S}(\mathbf{Q}, \mathbf{F}_{\text{attr}}). \qquad (10)$$

During inference, attribute-level logits can be aggregated to query-level predictions via $\mathbf{S}_{\text{query}}^{agg} = \mathcal{G}(\mathbf{S}_{\text{attr}}, \mathbf{M}_{map})$, where $\mathbf{M}_{map}$ is the hierarchical mapping matrix (defined in Sec. 3.2) and $\mathcal{G}(\cdot)$ aggregates attribute scores belonging to the same query. This enables users to query targets through either holistic expressions or flexible attribute compositions, with seamless conversion between both granularities.

### 3.3.4. MULTI-TASK LOSS

The total training objective is formulated as a weighted summation of semantic alignment and localization losses:

$$\mathcal{L}_{\text{total}} = \sum_{k \in \mathcal{K}} \lambda_k \mathcal{L}_k, \qquad (11)$$

where $\mathcal{K} = \{\text{query}, \text{attr}, \text{box}, \text{giou}, \text{fgl}, \text{ddf}\}$ denotes the set of loss components. Specifically, the localization losses ( $\mathcal{L}_{\text{box}}$, $\mathcal{L}_{\text{giou}}$, $\mathcal{L}_{\text{fgl}}$, $\mathcal{L}_{\text{ddf}}$ ) follow the standard configuration of the base architecture. For semantic alignment

($\mathcal{L}_{\text{query}}$ and $\mathcal{L}_{\text{attr}}$), we adapt the Matchability-Aware Loss (MAL) (Huang et al., 2025b) to optimize visual-language semantic alignment.

**Semantic Alignment via MAL.** MAL leverages IoU as a soft target to dynamically modulate supervision signals, replacing rigid binary $\{0, 1\}$ labels. This mechanism ensures that high semantic scores are assigned only to accurately localized objects. Given vision-language similarity logits $\mathbf{S}$ (computed in Eq. 10), we first compute alignment probabilities $p = \sigma(\mathbf{S})$ via sigmoid activation, then optimize these scores using the MAL objective:

$$\mathcal{L}_{\text{MAL}}(p, q, y) = \begin{cases} -[q^{\gamma} \log p + \bar{q}^{\gamma} \log \bar{p}], & y = 1, \\ -\alpha p^{\gamma} \log \bar{p}, & y = 0, \end{cases} \qquad (12)$$

where $\bar{q} = 1 - q$, $\bar{p} = 1 - p$, $q = \text{IoU}(b_i, b_j^*)$ is the soft quality target, $y \in \{0, 1\}$ is the binary label from the Unified Correspondence Matrices ($\mathbf{M}_Q$ or $\mathbf{M}_A$).

This adaptation bridges the gap between contrastive learning and dense object detection. Applying MAL to our decoupled multi-granular head, we compute alignment probabilities $p_{ij} = \sigma(\mathbf{S}_{\text{query}}[i, j])$ and $p_{ik} = \sigma(\mathbf{S}_{\text{attr}}[i, k])$ for queries and attributes respectively. The semantic alignment losses are then formulated as:

$$\mathcal{L}_{\text{query}} = \frac{1}{N_{\text{pos}}} \sum_{i,j} \mathcal{L}_{\text{MAL}}(p_{ij}, q_i, \mathbf{M}_Q[i, j]), \qquad (13)$$

$$\mathcal{L}_{\text{attr}} = \frac{1}{N_{\text{pos}}} \sum_{i,k} \mathcal{L}_{\text{MAL}}(p_{ik}, q_i, \mathbf{M}_A[i, k]), \qquad (14)$$

where $q_i = \text{IoU}(b_i, b_j^*)$ is the IoU target, $N_{\text{pos}}$ is the number of positive samples from the matcher, and $i, j, k$ index predicted boxes, queries, and attributes respectively.

## 4. Experiments

In this section, we first present datasets and implementations details in the experiments.

### 4.1. Datasets and Implementation Details

**Training Datasets.** Benefiting from our unified framework, we enable joint training across both paradigms. Specifically, we integrate the training sets from OVAD datasets (LAE-1M (Pan et al., 2025)) and RSVG datasets (OPT-RSVG (Li et al., 2024a), DIOR-RSVG (Zhan et al., 2023), AerialVG (Liu et al., 2025)), which we refer to as **OTA-Mix** throughout our experiments. Notably, RSVG data is reorganized via Image-Level Annotation Aggregation and enriched with fine-grained attributes through Attribute-Level Data Decomposition, as described in Sec. 3.

**Evaluation Datasets.** For OVAD evaluation, we assess the unified model on three benchmarks: DIOR, DOTAv2.0,

*Table 1.* Comparison with state-of-the-art methods on open-vocabulary aerial detection (OVAD) and remote sensing visual grounding (RSVG) benchmarks. Methods without specified pre-training data are trained on respective benchmark train sets. † indicates open-vocabulary detectors adapted for RSVG using the Naive Reformulation strategy described in Sec. 3.1.1.

| METHOD | PRE-TRAIN | FPS | OVAD BENCHMARK | | | RSVG BENCHMARK | | |
|---|---|---|---|---|---|---|---|---|
| | | | DIOR ($AP_{50}$) | DOTA (MAP) | LAE-80C (MAP) | OPT-RSVG (ACC@0.5) | DIOR-RSVG (ACC@0.5) | AERIALVG (ACC@0.5) |
| *Open Vocabulary Aerial Detection* | | | | | | | | |
| GLIP-T (CVPR'22) | LAE-1M | - | 82.8 | 43.0 | 16.5 | - | - | - |
| G-DINO-T (ECCV'24) | LAE-1M | 11 | 83.6 | 46.0 | 17.7 | - | - | - |
| LAE-DINO (AAAI'25) | LAE-1M | 10 | 85.5 | 46.8 | 20.2 | - | - | - |
| *Remote Sensing Visual Grounding* | | | | | | | | |
| TRANSVG (ICCV'21) | - | - | - | - | - | 70.0 | 72.4 | 11.5 |
| MGVLF (TGRS'23) | - | - | - | - | - | 72.2 | 76.8 | - |
| LPVA (TGRS'24) | - | 26 | - | - | - | 78.0 | 82.3 | - |
| AERIALVG (ICCV'25) | - | 7 | - | - | - | - | - | 50.0 |
| *Unified Detectors* | | | | | | | | |
| G-DINO-T† (ECCV'24) | OTA-MIX | 11 | 89.8 | 45.3 | 23.7 | 85.3 | 85.9 | 27.5 |
| LAE-DINO† (AAAI'25) | OTA-MIX | 10 | 89.0 | 46.2 | 23.3 | 84.9 | **86.7** | 35.7 |
| OTA-DET-S (OURS) | OTA-MIX | **34** | 91.9 | 48.0 | 29.2 | 85.0 | 82.7 | 49.2 |
| OTA-DET-M (OURS) | OTA-MIX | 32 | 91.8 | 49.4 | 28.8 | 85.2 | 83.8 | 51.7 |
| OTA-DET-L (OURS) | OTA-MIX | 28 | **93.1** | **52.0** | **31.0** | **86.5** | 85.1 | **53.9** |

and LAE-80C, following the evaluation protocol proposed by LAE-DINO (Pan et al., 2025). For RSVG evaluation, we conduct experiments on three benchmarks: OPT-RSVG, DIOR-RSVG, and AerialVG, using their respective test sets.

**Implementation Details.** For model initialization, the text encoder is initialized with pre-trained SigLIPv2 (Tschannen et al., 2025) (ViT-B-16-512). The image encoder is initialized with DINOv3-STAs (Huang et al., 2025a; Siméoni et al., 2025), following the same configuration as DEIMv2. The Multi-Task Loss weights are set to $\lambda = \{1.0, 1.0, 5.0, 2.0, 0.15, 1.5\}$ respectively. All other hyperparameters follow the default configuration of the base model. Pre-training with OTA-Mix is conducted on eight A800 GPUs. Unless otherwise specified, all other experiments are conducted on four RTX 4090 GPUs. The frames per second (FPS) are measured on a single RTX 4090 GPU with offline text encoding.

### 4.2. Evaluation Metrics

**OVAD Metrics.** For OVAD evaluation, we adopt standard object detection metrics following LAE-DINO (Pan et al., 2025). Specifically, we report mean Average Precision (mAP) across all categories and $AP_{50}$ at IoU threshold 0.5.

**RSVG Metrics.** For RSVG evaluation, following standard REC protocols, we adopt Top-1 Accuracy at IoU threshold 0.5 (Acc@0.5). To further assess fine-grained attribute understanding, we propose a stricter evaluation metric: *Attribute Alignment Accuracy* (Attr-Align@$\tau$). This metric requires predictions to satisfy both (i) correct localization under Acc@0.5, and (ii) average similarity between object

*Table 2.* Comparison on OVAD benchmarks with LAE-1M pre-training.

| METHOD | DIOR | DOTA | LAE-80C |
|---|---|---|---|
| GLIP-T | 82.8 | 43.0 | 16.5 |
| G-DINO-T | 83.6 | 46.0 | 17.7 |
| LAE-DINO | 85.5 | 46.8 | 20.2 |
| **OTA-DET-M** | **87.1** | **51.4** | **31.2** |

*Table 3.* Comparison on AerialVG.

| METHOD | Acc@0.5 |
|---|---|
| TRANSVG | 11.5 |
| G-DINO† | 29.4 |
| LAE-DINO† | 37.3 |
| AERIALVG | 50.0 |
| **OTA-Det-M** | **54.9** |

visual features and the referring expression's constituent attributes exceeding threshold $\tau$.

### 4.3. Performance on OVAD and RSVG Benchmarks

**Joint Training Results.** Table 1 presents comprehensive results across both OVAD and RSVG benchmarks. Unlike existing methods that address OVAD and RSVG separately, OTA-Det unifies both paradigms within a single architecture through joint training on OTA-Mix, achieving superior performance across both tasks while maintaining real-time efficiency. On OVAD benchmarks, compared to LAE-DINO, OTA-Det-L attains 93.1 $AP_{50}$ on DIOR (+7.6), 52.0 mAP on DOTA (+5.2), and 31.0 mAP on LAE-80C (+10.8). On

RSVG benchmarks, compared to specialized methods, OTA-Det-L achieves significant improvements: +8.5 Acc@0.5 on OPT-RSVG, +2.8 on DIOR-RSVG, and +3.9 on AerialVG. Moreover, OTA-Det-S maintains real-time inference at 34 FPS, 3× faster than LAE-DINO and 5× faster than AerialVG. Furthermore, we extend Grounding-DINO and LAE-DINO with the Naive Reformulation strategy (Sec. 3.1.1) to train on OTA-Mix for fair comparison. The results show that joint training enables these methods to handle both tasks, yet OTA-Det still maintains substantial advantages in both accuracy and efficiency, validating the effectiveness of our unified strategy and architectural design.

**Task-Specific Training Results.** To validate OTA-Det's effectiveness beyond joint training, we evaluate performance under task-specific settings. For OVAD evaluation, we train OTA-Det-M on LAE-1M following standard protocols. Table 2 shows that OTA-Det-M achieves state-of-the-art performance across all benchmarks, with particularly notable improvements on LAE-80C (+11.0 mAP). This substantial gain highlights OTA-Det's strong generalization capability for open-vocabulary detection. For RSVG evaluation, we assess performance on the challenging AerialVG benchmark, which features complex referring expressions with intricate spatial relationships. Table 3 shows that OTA-Det-M surpasses the previous best by 4.9 points. Notably, single-task training achieves slightly higher performance than joint training. This is expected since single-task models can fully optimize for one objective, while joint training requires balancing both OVAD and RSVG tasks. Nevertheless, the small performance gap demonstrates that our unified architecture effectively handles both tasks without significant degradation.

## 4.4. Ablation Studies

Table 4 presents ablation results on AerialVG using OTA-Det-M, where components are progressively removed from the full model. We analyze contributions along two aspects following our method presentation.

**Task Reformulation Strategy.** *(i) Naive Reformulation.* The last row of Table 4 (w/o Dense Align) represents the model trained with Naive Reformulation, which transforms RSVG into the joint classification-localization objective. While achieving competitive Acc@0.5 (54.67), the poor Attr-Align metrics (0.47% at $\tau$=0.7) reveal severe semantic pseudo-alignment. This indicates the model binds visual features to holistic sentences without comprehending constituent attributes. *(ii) Image-Level Annotation Aggregation.* Besides, Naive Reformulation also suffers from sparse supervision, limiting the model's discriminative capability. As shown in Fig. 3, without Image-Level Annotation Aggregation, the model produces numerous high-confidence false positives when processing multiple queries. After in-

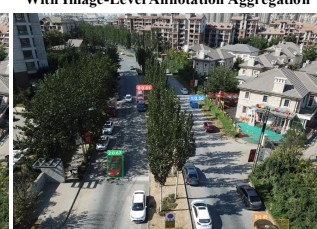

W/O Image-Level Annotation Aggregation    With Image-Level Annotation Aggregation

[0] A white van is parked on the roadside, with a blue truck parked below of it.
[1] A blue SUV is parked on the roadside, there is a red SUV parked at the bottom right and a white sedan at the top left.
[2] A red sedan is driving on the road, there is a white sedan parked at the bottom right.
[3] A black sedan is parked on the side of the road, there is a silver gray sedan parked at the top left.

*Figure 3.* Visualization of detection results with and without Image-Level Annotation Aggregation. Without aggregation (left), the model produces numerous false positives with high confidence due to sparse supervision. With aggregation (right), the model exhibits improved discriminative capability and accurately localizes the targets corresponding to each referring expression.

*Table 4.* Ablation study on key components of OTA-Det evaluated on the AerialVG benchmark.

| METHOD | STANDARD ↑ | ATTR-ALIGN ↑ | | |
|---|---|---|---|---|
| | ACC@0.5 | $\tau$=0.5 | $\tau$=0.6 | $\tau$=0.7 |
| OTA-DET-M | **54.94** | **44.23** | **34.36** | 21.96 |
| W/O IMG-AGG. | 54.56 | 43.21 | 33.90 | **22.74** |
| W/O DECOUPLED | 54.16 | 39.55 | 27.14 | 14.88 |
| W/O DENSE ALIGN | 54.67 | 14.36 | 3.81 | 0.47 |

corporating aggregation, which provides dense supervision through multiple query-target pairs per image, the model develops stronger discriminative ability. The quantitative metrics (second row of Table 4) confirm this trend.

**Dense Semantic Alignment Strategy.** *(i) Attribute-Level Alignment.* Comparing rows 3 and 4 in Table 4, the introduction of attribute-level alignment significantly mitigates semantic pseudo-alignment while maintaining comparable Acc@0.5. The improvements are substantial across all Attr-Align thresholds, with gains of +25.19% at $\tau$=0.5, +23.33% at $\tau$=0.6, and +14.41% at $\tau$=0.7. These results confirm that explicit fine-grained supervision enables the model to establish meaningful correspondence with individual semantic components rather than merely memorizing holistic expressions. *(ii) Decoupled Multi-Granular Head.* Comparing rows 2 and 3 in Table 4, our decoupled design yields consistent improvements across all Attr-Align metrics. By computing similarity logits independently for holistic queries and fine-grained attributes, this design prevents mutual interference between different semantic granularities, thereby enhancing fine-grained semantic understanding.

## 4.5. Qualitative Analysis

We provide qualitative visualizations in Fig. 4 and Fig. 5, demonstrating that OTA-Det successfully unifies OVAD and

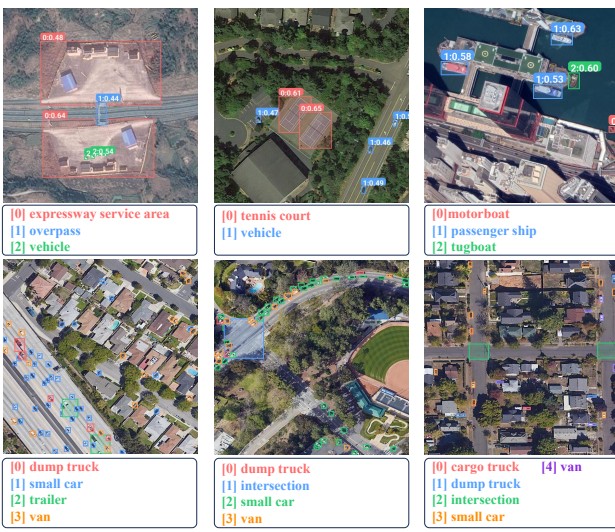

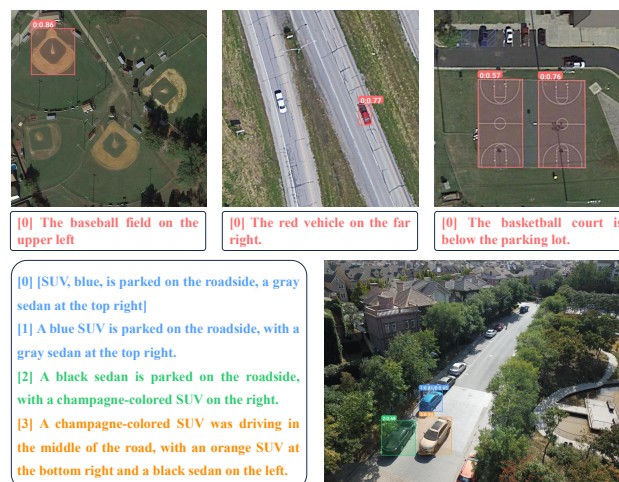

*Figure 4.* Qualitative results on open-vocabulary aerial detection. OTA-Det exhibits coarse category-level semantic understanding, supporting multi-query inference and one-to-many detection across varying scales and dense scenes. Each query ID (e.g., [0], [1]) corresponds to a different category name input simultaneously.

*Figure 5.* Qualitative results on visual grounding. Top: complex referring expressions with one-to-many detection. Bottom: multi-query inference with attribute-set interaction and multi-label association.

RSVG paradigms while introducing the extended capabilities claimed in Fig. 1.

**Open-Vocabulary Object Detection (Fig. 4).** OTA-Det exhibits category-level semantic understanding, supporting multi-query inference and one-to-many detection across varying scales and dense scenes. Fig. 4 demonstrates this across diverse aerial scenes, with the framework localizing all instances matching each category-level prompt.

**Visual Grounding (Fig. 5).** Top row demonstrates complex expression comprehension, satisfying RSVG's requirement of localizing targets based on holistic semantics. Benefiting from task reformulation (Sec. 3.1), OTA-Det enables confidence-based filtering for one-to-many detection, breaking the single-target limitation. For instance, the coarse-grained referring expression "The basketball court is below the parking lot" retrieves both basketball courts that satisfy this spatial relationship. Bottom row shows multi-query inference with 4 complex expressions processed simultaneously. Query [0] "[SUV, blue, is parked on the roadside, a gray sedan at the top right]" exemplifies attribute-set interaction where users input structured attributes with predictions aggregating via $M_{map}$ (Sec. 3.2.2). OTA-Det also handles multi-label association, as query [0] and query [1] correspond to the same target, validating the effectiveness of our dense semantic alignment strategy (Sec. 3.2).

## 5. Conclusion

Real-world aerial scene understanding demands simultaneous handling of multi-granular textual inputs and retrieval of all matching targets, while existing paradigms fail to satisfy these requirements when applied individually. In this paper, we propose OTA-Det, a unified framework that bridges OVAD and RSVG into a cohesive architecture to address this critical gap. Specifically, we first introduce a task reformulation strategy to address the fundamental discrepancies between these two paradigms, enabling joint training with dense supervision signals. Furthermore, we propose a dense semantic alignment strategy to establish explicit correspondence at multiple granularities, from holistic sentence representations to individual attribute components. This strategy not only mitigates the semantic pseudo-alignment problem but also introduces a novel attribute-set interaction mechanism that supports flexible compositional queries. To ensure real-time efficiency, OTA-Det builds upon the RT-DETR architecture, extending it from closed-set detection to open-text detection by introducing several highly efficient modules, achieving 34 FPS while exhibiting state-of-the-art performance on six benchmarks spanning both tasks.

## Acknowledgements

This work was supported in part by the National Natural Science Foundation of China under Grant 62401471, and in part by the 2024 Gusu Innovation and Entrepreneurship Leading Talents Program under Grant ZXL2024333.

## Impact Statement

This paper presents work whose goal is to advance the field of Machine Learning. There are many potential societal consequences of our work, none which we feel must be specifically highlighted here.

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

# Appendix Contents

# A. Related Work

## A.1. Open-Vocabulary Aerial Detection

Aerial object detection poses unique challenges, including predominantly small objects, diverse object variations, and complex background interference (Cheng et al., 2023; Tao et al., 2025; Zhou et al., 2025). Previous research (Li et al., 2020a; 2021; Huang et al., 2022; Du et al., 2023; Meethal et al., 2023) has primarily focused on closed-set detectors that improve model architectures to address these challenges and enhance small object detection accuracy. However, these detectors are constrained by training on fixed category sets and cannot identify novel classes, making them increasingly inadequate for real-world scenarios that require flexible category recognition. Inspired by successful approaches in natural images (Gu et al., 2022; Li et al., 2022; Liu et al., 2024; Cheng et al., 2024; Ren et al., 2024a; Wang et al., 2025), open-vocabulary aerial detection (OVAD) has emerged as a promising paradigm that establishes explicit correspondence between visual features and textual semantics through vision-language alignment, rather than simply regressing to fixed class indices (Han et al., 2024; Wei et al., 2024). Recent methods adopt different strategies to achieve open-vocabulary capability: DescReg (Zang et al., 2024) and CastDet (Li et al., 2024b) align proposal features with category semantics, while OVA-DETR (Wei et al., 2024) and OpenRSD (Huang et al., 2025c) exploit deep multimodal fusion to address the inherent challenges in aerial imagery. Besides, LAE-DINO (Pan et al., 2025) tackles the problem from the data perspective by constructing the large-scale LAE-1M dataset and establishing comprehensive OVAD benchmarks. Despite these advances, OVAD methods remain limited to coarse category-level semantics and lack the capability to comprehend sophisticated linguistic descriptions involving fine-grained attributes and complex spatial relationships.

## A.2. Remote Sensing Visual Grounding

Remote sensing visual grounding (RSVG) aims to locate objects in aerial images based on natural language descriptions. Compared to closed-set and open-vocabulary aerial detection, RSVG offers greater flexibility in textual interaction by processing arbitrary linguistic descriptions to identify corresponding targets, making it more suitable for practical applications (Xiao et al., 2024). Unlike multi-granular visual grounding tasks in natural images, such as Referring Expression Comprehension (REC) (Yu et al., 2016), Generalized Visual Grounding (He et al., 2023) and Phrase Localization (Plummer et al., 2015), RSVG remains in its early development stage. Existing datasets (Zhan et al., 2023; Li et al., 2024a; Zhou et al., 2024; Liu et al., 2025) predominantly focus on the REC task formulation, which inherently restricts the problem to single-target localization scenarios. Existing RSVG methods follow this single-target paradigm with varying architectural approaches. MGVLF (Zhan et al., 2023) focuses on multimodal fusion through transformer architectures to handle scale variations in remote sensing images. LPVA (Li et al., 2024a) explores earlier multimodal interaction within the backbone network, enabling the model to better locate targets corresponding to linguistic expressions. AerialVG (Liu et al., 2025) further investigates spatial relationship understanding in complex aerial scenes. However, these methods suffer from three fundamental limitations: (1) the REC formulation structurally restricts them to single-target scenarios, preventing multi-target detection; (2) semantic alignment is established only implicitly through localization supervision without explicit vision-language correspondence; and (3) existing evaluation metrics only assess whether models can locate targets based on holistic sentence semantics, neglecting whether models truly understand the fine-grained attribute semantics within complex expressions. These limitations hinder the practical deployment of RSVG systems in real-world applications that require comprehensive multi-target detection and fine-grained semantic understanding.

## A.3. Unified Exploration for Detection and Grounding

Object detection and visual grounding are both essential for real-world applications but are typically studied as separate tasks. Practical scenarios frequently require both capabilities simultaneously: detecting all relevant objects while comprehending linguistic descriptions at multiple granularities, from simple category names to complex referring expressions with spatial relationships. This necessity has motivated research efforts to unify detection and grounding within a single framework. GLIP (Li et al., 2022) pioneered this direction by reformulating object detection as a phrase grounding problem, enabling joint training on detection and phrase grounding datasets through a unified vision-language framework. Subsequent works (Liu et al., 2024; Ren et al., 2024b; Cheng et al., 2024; Wang et al., 2025) further advance this paradigm, demonstrating its effectiveness in natural image domains. However, phrase grounding and REC represent fundamentally different tasks: the former focuses on localizing objects corresponding to distinct noun phrases independently (e.g., separately locating "car" and "building" for the input "a red car near the building"), while the latter requires understanding the holistic semantics of entire sentences to identify specific targets defined by complex spatial or attribute relationships. A concurrent line of

*Table 5.* Statistics of datasets used in this work. For RSVG training sets, numbers in parentheses indicate caption counts after Image-Level Annotation Aggregation.

| DATASET | TASK | SPLIT | IMAGES | IMAGE-TEXT PAIRS | CATEGORIES | AVG. LENGTH |
|---|---|---|---|---|---|---|
| *Training Datasets (OTA-Mix)* | | | | | | |
| LAE-1M | OVAD | FULL | 88,017 | - | 1600 | - |
| DIOR-RSVG | RSVG | TRAIN | 14,730 | 26,991 (41,721) | 20 | 7.5 |
| OPT-RSVG | RSVG | TRAIN | 14,559 | 19,580 (34,139) | 14 | 10.1 |
| AERIALVG | RSVG | TRAIN | 4,876 | 37,788 (42,642) | - | 19.7 |
| *Evaluation Datasets* | | | | | | |
| DIOR | OVAD | VAL | 586 | - | 20 | - |
| DOTAv2.0 | OVAD | VAL | 874 | - | 18 | - |
| LAE-80C | OVAD | TEST | 3,592 | - | 80 | - |
| DIOR-RSVG | RSVG | TEST | 6,101 | 7,500 | 20 | 7.5 |
| OPT-RSVG | RSVG | TEST | 4,536 | 24,477 | 14 | 10.1 |
| AERIALVG | RSVG | TEST | 2,970 | 4,723 | - | 19.7 |

work, GREC (He et al., 2023), extends REC from single-target to multi-target localization, but remains a localization-only formulation within the grounding paradigm and is developed for natural images.

In the aerial domain, existing RSVG methods exclusively follow the REC paradigm, leaving the unification of detection and REC an open challenge. To address this gap, we propose OTA-Det, which unifies OVAD and RSVG to meet the practical detection demands in aerial imagery, enabling simultaneous handling of multi-granular textual inputs and retrieval of all matching targets. Our work differs from previous unification efforts in two key aspects: (1) we target the unification of OVAD and RSVG (REC paradigm) rather than phrase localization, addressing a fundamentally different task objective with larger semantic discrepancy; and (2) our motivation is specifically driven by bridging the gap between existing aerial paradigms and real-world application requirements, where OVAD is restricted to coarse category-level semantics with multi-target detection capability, whereas RSVG handles complex referring expressions but is limited to single-target scenarios.

# B. Details of Experiments and Additional Results

## B.1. Details of Datasets

Following the implementation details in Section 4.1, we train our unified OTA-Det model on OTA-Mix, which integrates both OVAD and RSVG datasets, and evaluate on benchmarks for both tasks.

**OVAD Datasets.** We adopt the OVAD benchmark established by LAE-DINO (Pan et al., 2025), which trains on LAE-1M and evaluates on DOTAv2.0, DIOR, and LAE-80C. *LAE-1M* is a large-scale open-vocabulary aerial detection dataset containing approximately 1 million annotation instances across 88,017 images with ~1,600 categories, comprising LAE-FOD (existing aerial detection data from (Xia et al., 2018; Li et al., 2020b; Sun et al., 2022a; Cheng et al., 2014; Long et al., 2017; Lam et al., 2018; Liu et al., 2017; Zhang & Deng, 2019)) and LAE-COD (annotations via the LAE-Label Engine from (Xia et al., 2017; Cheng et al., 2017; Yuan et al., 2022)). For evaluation, we use the validation sets of standard aerial object detection benchmarks: *DIOR* (586 images, 20 categories), *DOTAv2.0* (874 images, 18 categories), and *LAE-80C* (3,592 images, 80 categories), which is specifically designed for assessing open-vocabulary generalization capabilities.

**RSVG Datasets.** As shown in Table 5, we utilize existing visual grounding datasets covering remote sensing images (Zhan et al., 2023; Li et al., 2024a) and drone images (Liu et al., 2025), training on their training sets and evaluating on their test sets. *DIOR-RSVG* consists of 38,320 expressions across 17,402 images with 20 categories and an average expression length of 7.5 words. *OPT-RSVG* contains 48,952 image-query pairs across 25,452 images with 14 categories and an average expression length of 10.1 words. *AerialVG* is a recent drone visual grounding dataset featuring challenging referring expressions with intricate spatial relationships and an average expression length of 19.7 words, making it significantly more complex and challenging compared to the other two datasets. To enable dense supervision for discriminative learning, we reorganize RSVG training data via Image-Level Annotation Aggregation (described in Sec. 3), yielding 34,139 image-text pairs for OPT-RSVG, 41,721 for DIOR-RSVG, and 42,642 for AerialVG. Additionally, the training data is enriched with fine-grained attributes through Attribute-Level Data Decomposition to facilitate fine-grained semantic understanding. For

evaluation, the test sets contain 24,477 samples for OPT-RSVG, 7,500 samples for DIOR-RSVG, and 4,723 samples for AerialVG.

## B.2. Training Details

**Dense Semantic Alignment Strategy.** Existing visual-language alignment approaches typically adopt implicit alignment through contrastive learning and can be categorized into two strategies: (1) *Fine-grained token alignment* (e.g., Grounding DINO (Liu et al., 2024), GLIP (Li et al., 2022)) first tokenizes the input textual expression into multiple tokens, then aligns object visual features with each corresponding token. While this approach provides fine-grained correspondence, it suffers from two limitations: (i) it requires prior knowledge of which tokens correspond to each object in the data, and (ii) processing multiple tokens incurs prohibitive computational costs when handling long expressions, especially for subsequent multi-modal interaction modules, thereby reducing inference speed. (2) *Global alignment* (e.g., YOLO-World (Cheng et al., 2024)) aligns object visual features with a single global embedding that encodes the entire input expression via text encoders. Although computationally efficient, this strategy struggles to capture fine-grained semantic information in complex expressions through a single holistic representation, leading to spurious alignments (discussed in Sec. 1).

To address these limitations, we propose a Dense Semantic Alignment Strategy that explicitly aligns object visual features with both holistic query representations and fine-grained attributes. Specifically, we decompose textual expressions into constituent attributes and encode each attribute into a single global token. This design reduces the token burden for computational efficiency while preserving fine-grained semantic information for multi-granular understanding.

**Dynamic Text Sampling Strategy.** To enable efficient batch processing despite varying numbers of text prompts across samples, we employ a dynamic sampling strategy that unifies batch dimensions while enhancing training diversity.

*Sampling Configuration.* We set the maximum number of text prompts to 60, following design principles from prior work (Pan et al., 2025; Cheng et al., 2024). Based on dataset statistics showing that most referring expressions contain fewer than 10 distinct attributes, we set the maximum number of attributes per text to 10. These limits balance supervision density with computational efficiency. All sequences shorter than these limits are padded with empty tokens to ensure consistent tensor dimensions across batches.

*OVAD Sampling.* For each OVAD sample, we partition the vocabulary into positive classes $\mathcal{C}_{\text{pos}}$ (categories present in the image) and negative classes $\mathcal{C}_{\text{neg}}$ (remaining categories). We include all positive classes and randomly sample between 1 and $|\mathcal{C}_{\text{neg}}|$ negative classes to construct the text prompt. This negative sampling strategy introduces variability across training iterations, preventing the model from memorizing fixed category orderings.

*RSVG Sampling.* For RSVG samples, we randomly sample between 1 and $|\mathcal{C}_{\text{pos}}|$ referring expressions from the set of expressions associated with each sample. This random sampling encourages the model to handle diverse expression combinations and enhances robustness to inputs containing varying numbers of referring expressions.

*Attribute Sampling.* For category-based inputs (OVAD), we use the category name itself as the primary attribute. For referring expressions (RSVG), we sample from the fine-grained attribute set extracted through Attribute-Level Data Decomposition. For padded text positions, we also apply empty tokens to pad their corresponding attributes, ensuring uniform tensor shapes across the batch.

*Supervision Matrix Construction.* Based on the sampled texts and attributes, we construct the supervision matrices ($\mathbf{M}_Q$, $\mathbf{M}_A$, $\mathbf{M}_{\text{map}}$) as detailed in Sec. 3.2.2. During training, padding masks are applied to exclude padded positions from loss computation, ensuring that only valid text-visual correspondences contribute to gradient updates.

**Data Augmentation.** Our data augmentation strategy follows the base architecture (Huang et al., 2025a) with necessary adaptations for the unified training of OVAD and RSVG tasks. To maintain the semantic correspondence between visual content and referring expressions, we deliberately disable spatial augmentations (e.g., Mosaic, horizontal flipping) that could alter object positions or spatial relationships described in the text. Other preprocessing operations are retained to ensure training robustness while preserving grounding consistency.

*Table 6.* Sensitivity to the attribute loss weight $\lambda_{\text{attr}}$ on AerialVG.

| $\lambda_{\text{query}}$ | $\lambda_{\text{attr}}$ | Acc@0.5 | Attr@0.5 | Attr@0.6 | Attr@0.7 |
|---|---|---|---|---|---|
| 1.0 | 0.5 | 54.99 | 41.33 | 27.84 | 14.61 |
| 1.0 | 1.0 | 54.94 | 44.23 | 34.36 | 21.96 |
| 1.0 | 1.5 | 54.54 | 44.84 | 34.49 | 22.76 |
| 1.0 | 2.0 | 54.12 | 42.07 | 30.95 | 18.42 |

*Table 7.* Inference efficiency breakdown. The base architecture refers to the closed-set RT-DETR variant (DEIMv2-S) on which OTA-Det-S is built.

| Method | Type | FPS |
|---|---|---|
| G-DINO-T | open-text | 11 |
| LAE-DINO | open-text | 10 |
| Base Architecture | closed-set | 38 |
| OTA-Det-S (Ours) | open-text | 34 |

## C. Additional Experimental Analysis

### C.1. Sensitivity to Attribute Loss Weight

We adopt the same loss weight settings as the base architecture (Huang et al., 2025a) to avoid attributing performance gains to hyperparameter tuning. The only newly introduced loss term is $\mathcal{L}_{\text{attr}}$, controlled by $\lambda_{\text{attr}}$. Table 6 reports a sensitivity analysis on AerialVG. Performance remains stable for $\lambda_{\text{attr}} \in [1.0, 1.5]$ and drops at 2.0. We adopt $\lambda_{\text{attr}} = 1.0$ as the default to keep $\mathcal{L}_{\text{attr}}$ on equal footing with $\mathcal{L}_{\text{query}}$.

### C.2. Inference Efficiency Analysis

The inference efficiency of OTA-Det stems from two design choices. First, in extending the closed-set base detector to open-text detection, we introduce only the Decoupled Multi-Granular Head, which consists of lightweight linear projections. Second, offline text encoding decouples text computation from inference at deployment. Beyond these architectural choices, our core innovations (task reformulation and dense semantic alignment) operate at the problem formulation and supervision level, reshaping training signals without adding any inference-time module.

Table 7 reports a per-method FPS comparison on a single RTX 4090 GPU. OTA-Det incurs only a 4 FPS drop from the closed-set base architecture, confirming that our open-text extension introduces minimal overhead while preserving the base detector's speed advantage.

### C.3. Comparison with Remote Sensing VLMs

We further compare OTA-Det against recent remote sensing vision-language models (RS VLMs) on DIOR-RSVG. As shown in Table 8, OTA-Det-L achieves strong competitive performance on the same benchmark while maintaining real-time inference.

## D. Limitation Analysis

OTA-Det unifies OVAD and RSVG tasks into a cohesive architecture, enabling joint training on their datasets and achieving state-of-the-art performance on both tasks. However, there is still room for improvement in several aspects:

*(i) Data Scale.* In natural image domains, similar unified methods demonstrate that vision-language alignment is fundamentally a data-driven task. For instance, Grounding DINO v1.5 (Ren et al., 2024b) was trained on over 20 million grounding annotations, while DINO-X (Ren et al., 2024a) utilized over 100 million grounding annotations, both achieving strong semantic understanding performance. In contrast, OTA-Mix remains relatively limited, containing less than 0.1 million grounding annotations for training. Expanding the scale and diversity of aerial vision-language data represents a promising direction for future work.

*(ii) Attribute Semantic Guided Feature Enhancement.* OTA-Det aims to construct an efficient and concise unified architecture.

*Table 8.* Comparison with representative remote sensing VLMs on DIOR-RSVG.

| Method | DIOR-RSVG Acc@0.5 |
|---|---|
| VHM (Pang et al., 2025) (AAAI'25) | 56.17 |
| EarthGPT (Zhang et al., 2024) (TGRS'24) | 76.65 |
| GeoGround (Zhou et al., 2024) (arXiv'24) | 77.73 |
| OTA-Det-L (Ours) | **85.10** |

**Dense same-category targets**

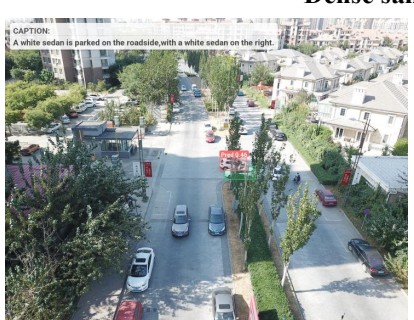 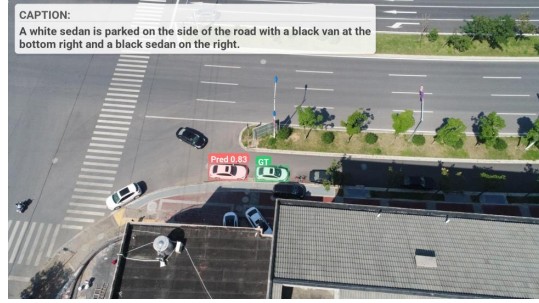

**Ambiguous spatial references**

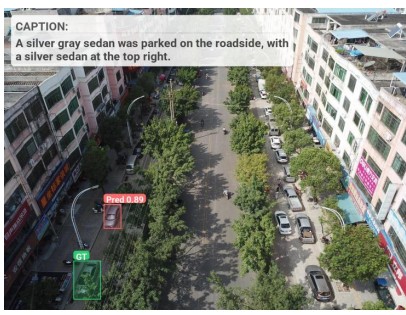 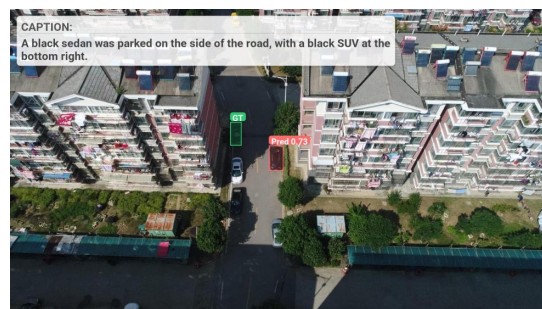

*Figure 6.* Failure cases on the AerialVG test set. Top: dense same-category targets where multiple similar instances cause prediction confusion. Bottom: ambiguous spatial references where the reference object is described similarly to the target. Green boxes denote ground truth (GT); red boxes denote predictions (Pred) with confidence scores.

Consequently, we did not explore complex multimodal interaction mechanisms that could leverage fine-grained attribute semantics for enhanced feature extraction. However, existing research (Pan et al., 2025; Liu et al., 2024; Wei et al., 2024) demonstrates that language-guided feature enhancement effectively addresses inherent challenges in aerial imagery, such as complex background interference and weak small-object feature representation. Given that our framework introduces fine-grained attribute semantics through the Dense Semantic Alignment Strategy, how to effectively utilize this information for deeper multimodal interaction, such as attribute-guided visual feature extraction or fine-grained cross-modal fusion, represents a promising research avenue.

*(iii) Failure Case Analysis.* We analyze failure cases from two perspectives.

*Test-set patterns* (Fig. 6). On AerialVG, the model fails most often in two scenarios. (a) *Dense same-category targets*: when several visually similar instances appear together, the model often selects a neighboring one instead of the true target. (b) *Ambiguous spatial references*: when the reference object resembles the target (e.g., "a white sedan..., with a white sedan at the top right"), the model struggles to distinguish target from reference.

*Attribute sensitivity* (Fig. 7). We probe the model with custom attribute sets in which only one attribute is altered (e.g., color: blue to black; yellow to white) while the others stay correct. The predicted similarity drops noticeably (0.90 to 0.69; 0.71 to 0.31), showing that the Decoupled Multi-Granular Head does capture attribute-level semantics. However, the wrong-attribute scores remain non-zero because the head averages scores across attributes, and a single mismatch is diluted by the remaining correct ones. Addressing this dilution through attribute-importance weighting is left for future work.

Despite these limitations, OTA-Det establishes a strong foundation for unified aerial detection and grounding at both the

**Correct --- Score: 0.90**        **Wrong --- Score: 0.69**

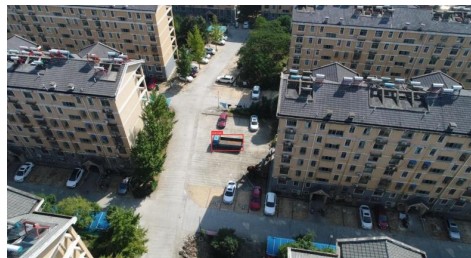 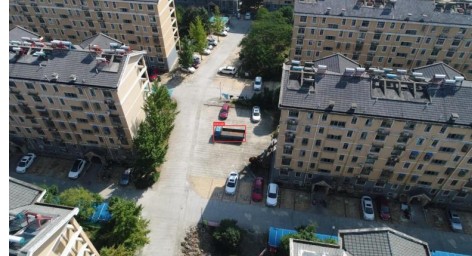

**Prompt**: [ 'blue', 'truck', 'parked in the vacant area', 'below the residential building with a red sedan above it']

**Prompt**: [ 'black', 'truck', 'parked in the vacant area', 'below the residential building with a red sedan above it']

**Correct --- Score: 0.71**        **Wrong --- Score: 0.31**

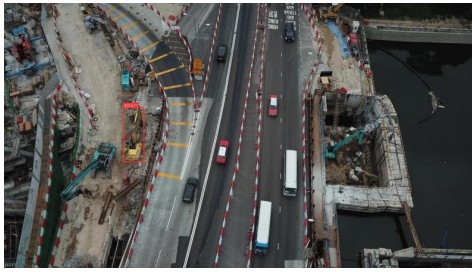 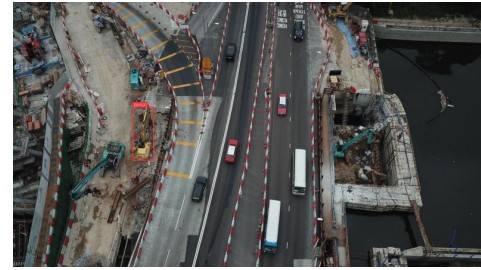

**Prompt**: ['yellow', 'excavator', 'parked on the side of the road', 'with a blue-green excavator at the bottom left']

**Prompt**: ['white', 'excavator', 'parked on the side of the road', 'with a blue-green excavator at the bottom left']

*Figure 7.* Attribute-level sensitivity. Each row shows the same target queried with two attribute sets that differ in only one attribute (highlighted in red). The score drop demonstrates that the model captures attribute-level semantics, while the residual non-zero score on the wrong attribute reveals a dilution effect from averaging.

task formulation and architectural levels, providing a solid basis for future research in this direction.

## E. Prompt for Attribute-Level Data Decomposition

Attribute-Level Data Decomposition is a one-time offline preprocessing step, decoupled from model training and inference. We employ a carefully designed prompt to guide an LLM in decomposing referring expressions into structured attributes. Given the simplicity of the task, we initially adopted GPT-OSS-120B (Agarwal et al., 2025) (MXFP4, fully open-source) more out of convenience than necessity, deployed on a single 80 GB GPU and processing each referring expression in approximately 1.5 seconds. To validate the reliability of the extracted attributes, three expert annotators independently verified 1,000 randomly sampled expressions per dataset (3,000 in total), achieving 100% agreement with the LLM outputs. We additionally tested GPT-OSS-20B, deployable on a single 16 GB consumer GPU, on 600 random samples, obtaining 100% agreement at approximately 1.4 seconds per expression. This confirms that the pipeline is reproducible at substantially lower compute without quality degradation. The complete prompt is presented below:

---

**Prompt for Attribute Extraction**

**Role:** You are a grounding-caption analyzer designed to extract target-centric attributes from visual grounding captions.

**Task:** Given a referring expression, identify the PRIMARY TARGET object and extract ALL its attributes as verbatim phrases directly from the caption.

**Critical Understanding:**

---

Before extracting attributes, understand that:
- The ENTIRE caption describes ONE single target object for visual grounding, not multiple separate targets.
- ALL words in the caption must be parsed; the entire expression serves to uniquely identify this one target.
- Other objects mentioned (e.g., cars, buildings, people) are REFERENCE OBJECTS that help locate the primary target through spatial relationships.
- These reference objects are NOT separate targets; they exist solely to describe WHERE or IN RELATION TO WHAT the target is located.

**Extraction Rules:**

Follow these rules strictly when extracting attributes:
1. *Parse completely:* Process EVERY word in the caption. Do not stop early; the full sentence contributes to grounding the target.
2. *Target-centric only:* Every extracted attribute must describe the target itself. Reference objects should appear only within `spatial_relation` attributes.
3. *Verbatim extraction:* The `description` field must be an exact substring from the caption. No paraphrasing, no added words, no synonyms.
4. *Complete spatial relations:* When a clause mentions other objects (e.g., "a white sedan in front of it"), extract it COMPLETELY as a `spatial_relation` attribute; do not break it apart.
5. *Maintain semantic coherence:* Keep semantically connected phrases together as single attributes. For example, "driving the left side onto the road" is ONE `state` attribute, not separate `position` and `environment` attributes.
6. *No hallucination:* Only extract what is explicitly stated. If uncertain about any aspect, omit it rather than inferring or adding information.
7. *Output format:* Return a single valid JSON object without markdown code fences.

**Attribute Aspects:**

The following aspects may be present in captions: `category`, `color`, `size`, `shape`, `material`, `texture`, `number`, `state`, `part`, `text`, `brand`, `activity`, `pose`, `status`, `position`, `orientation`, `spatial_relation`, `distance`, `environment`, `weather`, `time`, `context`, `purpose`, or any other observable property.

**Example:**

*Input Caption:*

> "A black van is driving the left side onto the straight road, a white sedan driving in front of it at the top right."

*Expected Output:*

```
{
  "primary_target": "van",
  "attributes": [
    {
      "aspect": "category",
      "description": "van",
      "caption_evidence": ["A black van"],
      "confidence": 1.0
    },
    {
      "aspect": "color",
      "description": "black",
      "caption_evidence": ["A black van"],
      "confidence": 1.0
    },
    {
      "aspect": "state",
      "description": "is driving the left side onto the straight road",
      "caption_evidence": ["is driving the left side onto..."],
      "confidence": 1.0
    },
```

```
        {
          "aspect": "spatial_relation",
          "description": "a white sedan driving in front of it at the top right",
          "caption_evidence": ["a white sedan driving in front of it..."],
          "confidence": 1.0
        }
      ],
      "analysis": "Target is 'van'. The phrase 'is driving the left side
                   onto the straight road' is kept as a complete action
                   describing the movement trajectory. The clause about
                   the white sedan represents a spatial reference."
    }
```

