# OpenReview forum: "Open-Text Aerial Detection: A Unified Framework For Aerial Visual Grounding And Detection"
_ICML.cc/2026/Conference — ICML 2026 regular_

### Official Review · Reviewer_AjpB · 2026-03-01

**Soundness:** 3
**Presentation:** 3
**Significance:** 2
**Originality:** 3
**Overall Recommendation:** 4
**Confidence:** 4

**Summary:**

This paper introduces a unified framework OTA-Det that bridges OVAD and RSVG into a single architecture. It first proposes task reformulation strategy to reformulate RSVG into a joint classification-localization task. Then, it designs a dense semantic alignment mechanism for multi-level supervision via attribute-level decomposition. Finally, a real-time unified architecture OTA-Det is built, demonstrating strong performance across six benchmarks spanning both OVAD and RSVG tasks.

**Compliance With Llm Reviewing Policy:**

Affirmed.

**Final Justification:**

I prefer to maintain my current score for the overall quality.

**Key Questions For Authors:**

1. In the experiments, drone VG dataset AerialVG is used, which has intrinsic domain gap and more complex positional annotations compared to other remote sensing/satellite datasets presented. How does OTA-Det handle domain shift or semantic drift between these datasets?
2. Closely related real-time open-vocabulary aerial detection work RT-OVAD [1] is not discussed. It is also recommended authors could include more comparisons with popular remote sensing VLMs that can perform multiple tasks including RSVG and OVAD.
3. There is a limit (e.g., max 60 queries, 10 attributes per text) imposed for efficiency. How would the model’s performance and computational cost change if the number of queries or attributes were substantially increased for much larger real-world or streaming query sets?

[1] RT-OVAD: Real-Time Open-Vocabulary Aerial Object Detection via Image-Text Collaboration

**Limitations:**

Limitations are discussed, it could be helpful to include potential societal impact as well.

**Strengths And Weaknesses:**

**Strengths**
1. The paper is technically sound. It directly addresses an important gap in the aerial scene understanding by enabling simultaneous multi-target detection and rich semantic grounding, which previously required separate or specialized models.
2. The paper is well organized, with clear explanations and an extensive discussion of related work, datasets, and benchmarking protocols.

**Weaknesses**

See Key Questions for details.

---

> ### Author Rebuttal · Authors · 2026-03-31
>
> We thank the reviewer for recognizing the technical soundness and clear presentation of our work. We appreciate the constructive feedback and address each question below.
>
> ---
>
> **Q1: Domain Shift between AerialVG and Other RS Datasets.**
>
> We thank the reviewer for this insightful question. We address this concern from three aspects:
>
> **(1) Analysis of cross-dataset domain gap.** We analyze the potential gap from two perspectives. In terms of resolution, RSVG datasets are primarily derived from DIOR at \~800×800, OVAD data is cropped to 1024×1024, and AerialVG contains images at approximately 1500×1000; these resolutions are relatively close and all inputs are further unified to 800×800 in OTA-Det. In terms of perspective, both satellite and drone imagery share a top-down viewpoint, and remote sensing images typically have high spatial resolution (\~0.3m), making the visual appearance of objects similar to those captured by drones, which naturally reduces the domain gap between these sources.
>
> **(2) Joint training provides domain robustness.** Enabled by our task-level unification, OTA-Det supports joint training across RSVG and OVAD datasets, encouraging the model to learn generalizable vision-language alignment rather than dataset-specific patterns. Image-Level Annotation Aggregation further reinforces this by providing dense supervision for the RSVG task.
>
> **(3) Empirical validation.** As discussed in Sec. 4.3, although joint training naturally involves trade-offs when balancing OVAD and RSVG across different domains, OTA-Det still achieves competitive performance on AerialVG (Table 1) compared to single-task training (Table 3), verifying that OTA-Det exhibits considerable domain robustness.
>
> Nevertheless, domain adaptation remains a valuable direction as the reviewer suggests. Incorporating explicit mechanisms (e.g., domain-specific prompts or adapters) could further bridge cross-domain gaps, which we will explore in future work.
>
> ---
>
> **Q2: Comparison with RT-OVAD and RS VLMs.**
>
> We thank the reviewer for this valuable suggestion. We address each recommendation below.
>
> **RT-OVAD.** We appreciate the reviewer pointing out this work. We note that RT-OVAD (arXiv) is a later version of OVA-DETR (Wei et al., 2024), which has already been discussed in our Related Work (Appendix A.1). It focuses on exploring image-text alignment to improve OVAD performance, while our work focuses on unifying OVAD and RSVG at the task formulation and training strategy level. We believe both works address different aspects of open-text aerial detection and are complementary in nature.
>
> **RS VLMs.** In our original submission, we did not include RS VLMs as baselines for two main reasons: (1) RS VLMs focus on general-purpose multi-task understanding (e.g., VQA, captioning, grounding), while OTA-Det targets efficient real-time unified detection for practical deployment; (2) RS VLMs typically operate at ~7B+ parameters for stronger performance, while our work prioritizes accuracy–efficiency balance.
>
> Following the reviewer's suggestion, we compare against representative RS VLMs on DIOR-RSVG:
>
> | Method                 | DIOR-RSVG Acc@0.5 |
> | ---------------------- | ----------------- |
> | VHM (AAAI 2025)        | 56.17             |
> | EarthGPT (TGRS 2024)   | 76.65             |
> | GeoGround (arXiv 2025) | 77.73             |
> | **OTA-Det-L (Ours)**   | **85.1**          |
>
> OTA-Det-L achieves strong competitive performance on the same benchmark while maintaining real-time inference.
>
> **Q3: Scalability of Query and Attribute Limits.**
>
> We thank the reviewer for this question.
>
> **Rationale for current settings.** The 60-query limit follows LAE-DINO, which demonstrated this ratio yields optimal training balance. The 10-attribute limit is determined by our statistical analysis of all RSVG datasets: after Attribute-Level Data Decomposition, decomposed attributes per expression never exceed 10, meaning this upper bound fully covers the actual data distribution without any truncation.
>
> **Minimal overhead from scaling.** Both parameters can be easily adjusted at inference time, and increasing them introduces only marginal computational overhead. Specifically, OTA-Det's image-text pipeline is fully decoupled: growth in the text branch does not affect the visual pipeline (encoder, deformable decoder), and only impacts two lightweight components: (1) Text Encoder encoding, which can be completely eliminated via offline pre-computation; (2) Decoupled Multi-Granular Head, computing image-text similarity via matrix multiplication. Let Q (object queries), d (embedding dimension), C (text queries), A (attributes per text). The head cost is:
>
> F_head = Q × d × C × (1 + A)
>
> Additional cost from increasing C by ΔC and A by ΔA:
>
> ΔF = Q × d × [ΔC × (1 + A + ΔA) + C × ΔA]
>
> This grows linearly with ΔC and ΔA, and the resulting computation is minimal and predictable, nearly negligible compared to the overall inference cost.
>
> ---

---

> > ### Author Rebuttal · Reviewer_AjpB · 2026-04-03
> >
> > Thank you for your response. My concerns have been addressed, and I will maintain my current score for the overall quality.

---

> > > ### Author Response · Authors · 2026-04-03
> > >
> > > We sincerely thank the reviewer for acknowledging that our responses have fully addressed the concerns. We greatly appreciate the constructive feedback throughout this review process, which has helped strengthen our manuscript.
> > >
> > > Best regards,
> > >
> > > The Authors

---

### Official Review · Reviewer_UMCU · 2026-03-06

**Soundness:** 2
**Presentation:** 3
**Significance:** 3
**Originality:** 2
**Overall Recommendation:** 4
**Confidence:** 2

**Summary:**

This paper introduces OTA-Det, a framework designed to bridge the gap between Open-Vocabulary Aerial Detection (OVAD) and Remote Sensing Visual Grounding (RSVG). By reformulating the RSVG task into a joint classification-localization problem and applying image-level annotation aggregation, the authors enable joint training across both task paradigms. To support fine-grained understanding, the method employs an LLM to decompose complex expressions into distinct attributes, constructing decoupled supervision matrices. Empirical results show that OTA-Det achieves strong performance on multiple aerial detection and grounding benchmarks while maintaining high inference speeds.

**Compliance With Llm Reviewing Policy:**

Affirmed.

**Final Justification:**

Regarding the online latency, while the results are now provided, they do not exhibit a substantial edge over other approaches. Furthermore, even though the failure cases are discussed in the rebuttal, corresponding qualitative visualizations are lacking. I would advise the authors to supplement the paper with more qualitative results to better support their method. Given these considerations, I prefer to keep my current rating of Weak Accept.

**Key Questions For Authors:**

please see weakness.

**Limitations:**

yes

**Strengths And Weaknesses:**

**Strengths**
- The paper identifies a highly practical and relevant problem: bridging the gap between single-target visual grounding and multi-target open-vocabulary detection in aerial imagery.
- The proposed Image-Level Annotation Aggregation is a clever and effective data-centric solution to align the structurally different supervision signals of OVAD and RSVG.
- The framework achieves strong empirical results across six distinct benchmarks, demonstrating impressive versatility across multi-granular inputs.
- The architecture is highly efficient, and the reported 34 FPS makes the method viable for real-world UAV deployment.

**Weaknesses**
- One of the highlights of this work is its real-time capability. However, the 34 FPS reported for OTA-Det-S is achieved with "offline text encoding". In many real-world applications where a user types a new complex query dynamically, the text must be encoded online. The omission of the end-to-end latency (including the SigLIPv2 encoding time) makes the efficiency claims slightly overstated and less reflective of dynamic use cases.
- While Figure 4 and Figure 5 show excellent qualitative successes in dense scenes, the paper completely omits visualizations or discussions of failure modes. It is critical to understand under what conditions the decoupled multi-granular head binds attributes to the wrong objects, especially in highly cluttered aerial environments with overlapping spatial references.
- The paper effectively transforms the problem into a multi-label classification scenario. However, it does not compare the unified framework against any adapted multi-label zero-shot detection models from the natural image domain. Including such a baseline would better contextualize whether the gains come from the aerial-specific adaptations or simply the multi-label contrastive formulation.

---

> ### Author Rebuttal · Authors · 2026-03-31
>
> We thank the reviewer for recognizing the practical relevance and strong empirical results of our work. We appreciate the constructive feedback and address each concern below.
>
> ---
>
> **W1: End-to-End Latency with Online Text Encoding.**
>
> We thank the reviewer for this practical consideration. We note that offline text encoding is well-aligned with typical UAV deployment, where text queries are pre-defined and do not change frequently during flight. Nonetheless, OTA-Det also supports online text encoding for dynamic queries. Following the reviewer's suggestion, we conduct additional latency measurements with SigLIPv2-B/16 running online. Even under this setting, OTA-Det achieves 16 FPS, maintaining significant advantages over existing methods (e.g., G-DINO/LAE-DINO at ~10 FPS) in both performance and speed.
>
> ---
>
> **W2: Failure Mode Visualization.**
>
> We thank the reviewer for this suggestion. We conduct failure analysis from two perspectives.
>
> **(1) Test set failure cases.** We identify two common failure patterns on AerialVG: (i) **Dense same-category targets**: when multiple visually similar objects co-exist, the model is prone to selecting the wrong instance. (ii) **Ambiguous spatial references**: when the reference object is described similarly to the target (e.g., "a white sedan ..., with a white sedan at the top right"), the model struggles to distinguish target from reference.
>
> **(2) Custom query verification.** To verify the reviewer's concern about attribute-object binding errors, we tested with custom-designed attribute sets. We found that when one attribute in the query does not match the target (e.g., incorrect color while category, state, and spatial relation are correct), the model may still retrieve the wrong object with high confidence. This is because the attribute matching score is averaged across all attributes, so a single mismatch is diluted by high scores on the remaining ones.
>
> Nonetheless, the experimental results validate that our model achieves its core objective of unifying OVAD and RSVG. The observed failure modes motivate future work: scaling up training data with more complex multi-object scenes and finer-grained textual annotations to establish stronger, more fine-grained vision-language understanding. We will include failure visualizations in the revised manuscript.
>
> ---
>
> **W3: Comparison with Multi-Label Zero-Shot Detection from Natural Image Domain.**
>
> We thank the reviewer for this suggestion. We explain below why direct comparison with multi-label zero-shot detection methods from the natural image domain is not applicable.
>
> **(1) Different task paradigms.**
>
> **From the object-level perspective**, the two approaches differ fundamentally in how attributes are used:
>
> - **Multi-label zero-shot detection** (e.g., Open-vocabulary Attribute Detection [Bravo et al., CVPR 2023]) first detects objects in an open-vocabulary manner and then determines which attributes are present for each detected object from a set of attribute candidates. Attributes are evaluated independently for each object, and each attribute is treated as an independent binary classification target.
> - **OTA-Det**, in contrast, aims to localize the target(s) corresponding to the whole expression or an attribute set. Specifically, the model uses the text query to select the target object(s) from all detected candidates. A key difference is that our attributes appear as a bound set tied to each caption, rather than independent classification targets. During training, the caption-attribute-target correspondence enables multi-granular understanding. During inference, this design allows the model to jointly leverage multiple attributes for more flexible and accurate detection.
>
> **From the task paradigm perspective**, to our knowledge, no existing method unifies detection and referring expression comprehension (REC). The most relevant works are GLIP and Grounding DINO, which unify detection with phrase grounding. However, phrase grounding and REC are fundamentally different tasks: the former detects all targets corresponding to all noun phrases within a sentence, treating each noun phrase as an independent query, while the latter requires understanding holistic sentence semantics to identify a specific target defined by complex spatial or attribute relationships (as discussed in Appendix A.3). Therefore, neither existing multi-label zero-shot detection methods nor existing unified detection-grounding methods can be directly and fairly compared.
>
> **(2) Representative baselines already included.** Despite these differences, we still include Grounding DINO, one of the most representative unified methods from the natural image domain, as a baseline. We adapt it using our Naive Reformulation strategy (Sec. 3.1.1) so that it can be trained on the identical OTA-Mix data alongside OTA-Det. Results demonstrate that OTA-Det achieves superior performance in both detection accuracy and efficiency.
>
> ---

---

> > ### Author Rebuttal · Reviewer_UMCU · 2026-04-03
> >
> > Thanks to the authors for their detailed response. Regarding the online latency, while the results are now provided, they do not exhibit a substantial edge over other approaches. Furthermore, even though the failure cases are discussed in the rebuttal, corresponding qualitative visualizations are lacking. I would advise the authors to supplement the paper with more qualitative results to better support their method. Given these considerations, I prefer to keep my current rating of Weak Accept.

---

> > > ### Author Response · Authors · 2026-04-04
> > >
> > > We thank the reviewer for the continued engagement and the positive assessment of our work.
> > >
> > > We apologize for not providing failure case visualizations directly in our initial response. We have now prepared visualizations for the two failure analyses identified in our rebuttal: (1) test set failure cases including dense same-category targets and ambiguous spatial references: [Visualization 1](https://anonymous.4open.science/r/ICML-Rebuttal-Repo-7D86/Fail_Case_Vis_1.png); (2) custom query verification for attribute-object binding errors: [Visualization 2](https://anonymous.4open.science/r/ICML-Rebuttal-Repo-7D86/attribute-object_binding_errors.png). We will incorporate these into the revised manuscript.

---

### Official Review · Reviewer_4LoK · 2026-03-10

**Soundness:** 2
**Presentation:** 2
**Significance:** 2
**Originality:** 2
**Overall Recommendation:** 2
**Confidence:** 4

**Summary:**

This paper proposes a unified framework that combines the multi-object detection capability of Open-Vocabulary Aerial Detection (OVAD) with the fine-grained semantic understanding of Remote Sensing Visual Grounding (RSVG). The authors introduce two main contributions: (1) a task reformulation strategy that unifies task objectives to enable joint training across both paradigms, and (2) a dense semantic alignment strategy that establishes explicit visual-semantic correspondence at multiple granularities. The framework achieved real-time inference at 34 FPS.

**Compliance With Llm Reviewing Policy:**

Affirmed.

**Final Justification:**

The rebuttal improves clarity and partially addresses my concerns. However, the quantitative evidence and analysis supporting the multi-object semantic capability remain limited, and the efficiency is difficult to attribute as a primary contribution of the proposed method. That said, the idea of unifying semantic understanding with detection is interesting and has potential for future work.

**Key Questions For Authors:**

1. Could the authors provide evaluation results on a benchmark that simultaneously requires fine-grained text queries and multi-object detection ground truth?
2. Since the proposed method is built upon RT-DETR, it is unclear whether the reported 34 FPS efficiency is primarily due to the proposed method or inherited from the underlying RT-DETR architecture. Could the authors provide a detailed comparison with the original RT-DETR, including inference latency and training time (or a per-module latency breakdown), to clarify the source of the efficiency gains?
3. The Attribute-Level Data Decomposition relies on GPT-OSS-120B over the full training set. Could the authors report the total number of samples processed, the inference time, and the associated training and fine-tuning costs?

**Limitations:**

No. The authors are encouraged to discuss the reproducibility concerns associated with GPT-OSS-120B and the lack of a unified evaluation benchmark as limitations of the proposed framework.

**Strengths And Weaknesses:**

Strengths
1. Practical motivation.
The paper addresses a genuine need in aerial image-based detection by handling fine-grained semantic text queries for aerial scene understanding.
2. Image-Level Annotation Aggregation.
Reorganizing sparse sentence-level triplets into dense image-level query sets is a simple yet effective approach that directly addresses the supervision density gap between OVAD and RSVG training data.

Weaknesses
1. Insufficient evaluation for the paper's defined objective.
The paper claims to unify fine-grained semantic understanding with multi-object detection, yet no evaluation benchmark simultaneously requires both capabilities.
2. Unaddressed overlap with GREC (Generalized Referring Expression Comprehension).
The proposed unified capability for multi-target detection with complex referring expressions has already been formulated in GREC. The paper does not justify why combining OVAD and RSVG is necessary beyond extending GREC to the aerial domain, which raises questions about the originality of the problem formulation.
3. The contribution of the FPS result is unclear.
The paper attributes 34 FPS to "several highly efficient modules" without explicitly identifying them. It is unclear whether the efficiency originates from the RT-DETR base architecture or from this work's contributions. A component-level latency breakdown is necessary to support this claim.
4. Lack of computational cost for the Attribute-Level Data Decomposition.
The Attribute-Level Data Decomposition relies on GPT-OSS-120B, yet accuracy was validated on only 1,000 samples per dataset, representing as little as 2.6% of training data, without inter-annotator agreement metrics. The inference time and cost of running GPT-OSS-120B over the full training set are also not reported, raising reproducibility concerns.

---

> ### Author Rebuttal · Authors · 2026-03-31
>
> Thank you for your valuable time and constructive feedback! We hope to address your primary concerns below.
>
> ---
>
> **W1 & Q1: Evaluation on Unified Benchmark.**
>
> Thank you for raising this point. Current aerial datasets are constructed independently for each paradigm: RSVG provides fine-grained semantic descriptions but follows the REC formulation with single-target annotations, while OVAD supports multi-object detection but is limited to category-level semantics. This paradigm-specific construction is precisely the gap our work aims to bridge through task reformulation.
>
> **Our evaluation already covers both capabilities.** Through task unification and joint training, a single OTA-Det model acquires both fine-grained understanding and multi-object detection capabilities. This is validated across six benchmarks (Table 1): OVAD benchmarks verify multi-object detection, RSVG benchmarks verify fine-grained semantic understanding, and the qualitative results (Fig. 4, 5) further demonstrate these capabilities working together.
>
> **Future direction.** We fully agree that constructing a unified benchmark is a valuable independent research direction, which is ongoing work on our end, but it is beyond the scope of this paper.
>
> ---
>
> **W2: Relationship with GREC.**
>
> We thank the reviewer for pointing out GREC. We will include a discussion in the revised related work. While GREC makes a valuable contribution to the REC community, we clarify the fundamental differences across three dimensions:
>
> **Task.** GREC extends REC from single-target to multi-target localization, but remains within the Grounding paradigm with localization-only supervision. OTA-Det operates at a different level: it unifies the Detection paradigm (explicit contrastive classification + localization) with the Grounding paradigm through task reformulation, which is a cross-paradigm unification rather than an intra-paradigm extension.
>
> **Capability.** Due to the lack of classification, GREC cannot support Multi-Query Inference (as analyzed in Fig. 1): when multiple prompts are queried simultaneously, GREC lacks classification capability and cannot assign each detection to the correct query. OTA-Det inherits capabilities from both paradigms and further enables Attribute-Set Interaction, Multi-Label Association, and Real-Time Inference, which cannot be achieved within the GREC framework.
>
> **Domain.** GREC is designed for natural images, and OTA-Det is specifically designed to address aerial-specific demands for practical UAV deployment.
>
> ---
>
> **W3 & Q2: FPS Contribution and Latency Breakdown.**
>
> We thank the reviewer for this question. The inference efficiency of OTA-Det stems from two aspects:
>
> **(1)** OTA-Det adopts the lightweight RT-DETR as its base architecture.
>
> **(2)** To ensure the inference speed advantage is fully inherited: (i) to extend it from closed-set detection to open-text detection, we only introduce several efficient Decoupled Multi-Granular Heads (consisting of linear projections); (ii) offline text encoding is leveraged to decouple text computation from inference.
>
> In addition, our core contributions (task reformulation and dense semantic alignment) operate at the problem formulation and supervision level, reshaping training signals rather than introducing additional inference-time modules. This ultimately enables OTA-Det to achieve real-time inference while delivering SOTA performance on six benchmarks spanning both OVAD and RSVG tasks.
>
> **W4 & Q3: Cost and Reliability of Attribute Decomposition.**
>
> We thank the reviewer for this concern. We would like to clarify that Attribute-Level Data Decomposition is a lightweight structured verbatim extraction task that identifies exact substrings from captions and assigns corresponding attribute types. Due to this simplicity, the LLM requires no fine-tuning or adaptation; only prompt design is needed, and thus the training cost is zero. We address the remaining concerns from three aspects below.
>
> **(1) Computational cost.** The decomposition is a one-time offline step, fully decoupled from model training and inference. We process 84,359 expressions across three datasets. Using GPT-OSS-120B (MXFP4, fully open-source) on a single 80GB GPU, the pipeline achieves ~1.5 sec/sample, completing the full dataset in ~36 GPU-hours. In other words, for a given task, the preparation cost is only seconds-level.
>
> **(2) Validation.** Three expert annotators independently verified 1,000 samples per dataset (3,000 total, ~3.6%), achieving **100% agreement** with LLM outputs.
>
> **(3) Model flexibility.** Given the task simplicity, our initial choice of GPT-OSS-120B was not driven by necessity but by convenience. We additionally tested GPT-OSS-20B (deployable on a single consumer GPU with 16GB VRAM) on 600 samples, achieving ~1.4 sec/sample with 100% accuracy, confirming that smaller models are equally capable and the full pipeline is reproducible with significantly lower compute.
>
> ---

---

> > ### Author Rebuttal · Reviewer_4LoK · 2026-04-02
> >
> > Thank you for the clarifications. If I understand correctly, the method is intended to unify fine-grained semantic understanding with multi-object detection within a single framework. I still have two follow-up questions regarding the main claims:
> >
> > Unified capability (within existing datasets).
> > From my understanding, the goal is to handle multi-object detection with rich semantic queries. However, the evaluation is conducted separately on OVAD and RSVG benchmarks. Do the current datasets actually contain examples that require both at the same time (e.g., complex expressions referring to multiple valid targets)? If so, could you provide concrete examples or analysis to support this?
> >
> > Efficiency (FPS).
> > The paper reports real-time performance (34 FPS), but since OTA-Det is built on RT-DETR, it is not entirely clear to me whether this efficiency comes from the proposed method itself or is largely inherited from the base architecture. Could you clarify this with a direct comparison or a latency breakdown?

---

> > > ### Author Response · Authors · 2026-04-03
> > >
> > > Q1: We thank the reviewer for the continued engagement and apologize for the lack of clarity in our initial response. We also appreciate the rebuttal mechanism that allows us to have this thorough exchange.
> > >
> > > First, we confirm that the reviewer's understanding of our goal is correct. We provide the following clarifications and hope they address the reviewer's concern.
> > >
> > > **(1) The limitation of current RSVG paradigm motivates our work.** Current RSVG follows the REC paradigm, where each caption corresponds to only a single target. However, **within current RSVG datasets**, many referring expressions naturally correspond to multiple valid targets in the image. As shown in https://anonymous.4open.science/r/ICML-Rebuttal-Repo-7D86/picture-1.png, these are **original queries from the dataset** should retrieve multiple matching targets in the image, yet only a single target is annotated due to the paradigm constraint. This observation was a key motivation for our work: **breaking the single-target limitation of the existing RSVG paradigm**.
> > >
> > > **(2) Our unified formulation enables multi-target output for complex expressions.** Original RSVG is a pure localization task (Eq. 3) that outputs only a single bounding box without any classification confidence, structurally limiting it to single-target output. Our Task Reformulation (Sec. 3.1) restructures RSVG into a joint classification-localization task (Eq. 5), introducing explicit contrastive learning supervision that produces a semantic similarity score $s_i \in [0,1]$ for each query-object pair. With this score, the model can naturally output multiple matching targets for a single complex expression — all targets with $s_i > \theta$ are retrieved. The visualization analysis in Fig. 5 also demonstrates that our model can effectively handle the scenario of complex expressions referring to multiple valid targets.
> > >
> > > **(3) This paradigm is gaining increasing attention from the community.** The scenario where complex expressions correspond to multiple valid targets is more aligned with real-world applications and has attracted growing community interest. For example, two recent works on **arXiv** — RefDrone [1] and MI-OAD [2] — have constructed datasets specifically targeting this setting (visualizations: [RefDrone](https://anonymous.4open.science/r/ICML-Rebuttal-Repo-7D86/refdrone_main_fig.png), [MI-OAD](https://anonymous.4open.science/r/ICML-Rebuttal-Repo-7D86/MI-OAD.jpg)). While these datasets are still under peer review, they have already received citations and positive reception (e.g., [OpenReview](https://openreview.net/forum?id=ZHc2Psv82y)). Once formally published, they will provide strong data support and serve as ideal evaluation benchmarks for our framework.
> > >
> > > [1] Sun Z, Liu Y, Su Z, et al. RefDrone: A Challenging Benchmark for Referring Expression Comprehension in Drone Scenes. arXiv preprint arXiv:2502.00392, 2025.
> > >
> > > [2] Wei G, Liu Y, Yuan X, et al. From Word to Sentence: A Large-Scale Multi-Instance Dataset for Open-Set Aerial Detection. arXiv e-prints, 2025: arXiv:2505.03334.
> > >
> > > ------
> > >
> > > Q2: We thank the reviewer for the continued engagement and apologize for the lack of clarity in our initial response. As the reviewer correctly identified, the real-time inference of OTA-Det is primarily inherited from the base architecture, **and our design contribution lies in ensuring that the speed advantage is fully preserved during the extension from closed-set detection to open-text detection**.
> > >
> > > Specifically, we maintain efficiency through the following design choices: (i) the Decoupled Multi-Granular Head consists only of lightweight linear projections; (ii) offline text encoding is leveraged to decouple text computation from inference. We provide the direct comparison as requested:
> > >
> > > | Model                 | Type           | FPS    |
> > > | --------------------- | -------------- | ------ |
> > > | G-DINO-T              | open-text      | 11     |
> > > | LAE-DINO              | open-text      | 10     |
> > > | **Base Architecture** | **closed-set** | **38** |
> > > | **OTA-Det-S (ours)**  | **open-text**  | **34** |
> > >
> > > The drop from the base architecture to OTA-Det is only 4 FPS, confirming that our extension introduces minimal overhead. Furthermore, our core innovations (task reformulation and dense semantic alignment) operate at the problem formulation and supervision level without introducing any additional inference-time modules, thereby preserving the base architecture's speed advantage within the open-text unified framework. We will include this comparison in the revised manuscript.

---

### Official Review · Reviewer_VRGf · 2026-03-13

**Soundness:** 2
**Presentation:** 2
**Significance:** 2
**Originality:** 2
**Overall Recommendation:** 3
**Confidence:** 4

**Summary:**

This paper proposes OTA-Det, a unified framework for Open-Text Aerial Detection, aiming to bridge the open-vocabulary aerial detection and remote sensing visual grounding. To unify the two tasks, the paper introduces key components such as Task Reformulation Strategy, Dense Semantic Alignment Strategy and Decoupled Multi-Granular Head. Experiments for the two tasks validate the effectiveness.

**Compliance With Llm Reviewing Policy:**

Affirmed.

**Key Questions For Authors:**

1) The key components of the proposed framework are mostly based on existing techniques, such as the common-used visual-text alignment, attribute decomposition and others. The authors need to clarify the novelty of the method.
2) The ablation study on the hyperparameters such as the weights in multitask losses, the effects of components such as Dense Semantic Alignment Strategy is required.
3) In Table 1, it is important to present visual encoder as well as text encoder. Since this work employs the Dinov3 visual encoder and Siglip2 text encoder, this raises the issue of fair comparison.
4) What about the robustness to incorrect attribute parsing?

**Limitations:**

Yes

**Strengths And Weaknesses:**

strengths
1. The attempt to unify OVAD and RSVG into a single unified framework is interesting.
2. Extensive experiments are conducted across several datasets.
3. The efficiency is good for practical deployment.

weakness
1. The experiment results need more proof for fair comparison.
2. The ablation study is required to illustrate to impacts of different designed components, especially for the two unified tasks.

---

> ### Author Rebuttal · Authors · 2026-03-31
>
> We greatly appreciate your detailed feedback and insightful comments. To address your concerns, we provide point-to-point responses below:
>
> ---
>
> **Q1: Novelty Clarification**
>
> We thank the reviewer for this question. We acknowledge that certain building blocks draw on existing ideas. However, unifying OVAD and RSVG poses unique challenges no prior work has addressed, requiring problem-specific redesign. As cross-attention is shared across DETR, Grounding DINO and GLIP yet each achieves novelty by adapting it to a distinct problem, **OTA-Det's novelty lies at the problem formulation and strategy level**: bridging two fundamentally different paradigms into a real-time unified framework.
>
> **(1) Problem-Level Contribution.** We formally identify two fundamental OVAD-RSVG discrepancies: task objective and semantic alignment mechanism, directly motivating subsequent design.
>
> **(2) Task Reformulation.** To bridge these discrepancies, we reformulate RSVG from pure localization into joint classification-localization, and propose Image-Level Annotation Aggregation to restructure data from sentence-level to image-level, enabling dense supervision. This unifies both tasks in task definition and supervision form.
>
> **(3) Dense Semantic Alignment.** During unification, we further discover the semantic pseudo-alignment problem. To address this, we propose Dense Semantic Alignment Strategy with fine-grained attribute-level supervision, enabling explicit alignment at multiple granularities from holistic expressions to individual attributes.
>
> **(4) Unified Architecture.** Building on these strategies, we construct the first unified architecture bridging OVAD and RSVG, inheriting complementary strengths of both tasks, achieving SOTA on six benchmarks at 34 FPS.
>
> ---
>
> **Q2 & W2: Ablation Study and Hyperparameter Sensitivity**
>
> We thank the reviewer for this suggestion.
>
> **(1) Component Ablation.** Table 4 provides systematic ablation by progressively removing each proposed component, demonstrating individual contributions.
>
> **(2) Unified Tasks Ablation.** The effect of joint training can be observed by comparing Table 1 and Table 2/3, with analysis at the end of Section 4.3.
>
> **(3) Loss Weight Sensitivity.** To avoid performance gains being attributed to hyperparameter tuning, we adopt the same loss weight settings with prior works (G-DINO, LAE-DINO, DEIMv2). Following the reviewer's suggestion, we additionally conduct sensitivity analysis on our extra introduced λ_attr:
>
> | λ_query | λ_attr | Acc@0.5 | Attr-Align@0.5 | Attr-Align@0.6 | Attr-Align@0.7 |
> | ------- | ------ | ------- | -------------- | -------------- | -------------- |
> | 1.0     | 0.5    | 54.99   | 41.33          | 27.84          | 14.61          |
> | 1.0     | 1.0    | 54.94   | 44.23          | 34.36          | 21.96          |
> | 1.0     | 1.5    | 54.54   | 44.84          | 34.49          | 22.76          |
> | 1.0     | 2.0    | 54.12   | 42.07          | 30.95          | 18.42          |
>
> Performance remains stable across λ\_attr∈[0.5, 1.5], indicating low sensitivity. We select λ_attr=1.0 for balanced trade-off between localization and attribute alignment.
>
> ---
>
> **Q3 & W1: Fair Comparison and Encoder Details**
>
> We thank the reviewer and will add backbone details for all methods in the revised Table 1.
>
> Beyond this, we would like to point out that an inherent asymmetry exists in the initial model conditions: methods like G-DINO and LAE-DINO are pretrained on massive natural-image grounding data (Objects365, GoldG, Cap4M), establishing strong vision-language alignment across all parameters, while OTA-Det uses stronger unimodal backbones (DINOv3 + SigLIPv2) but learns vision-language alignment entirely from scratch using only remote sensing data. Therefore, we fine-tune these methods on identical data to ensure fair comparison, and OTA-Det consistently outperforms across all six benchmarks.
>
> ---
>
> **Q4: Robustness to Incorrect Attribute Parsing.**
>
> We thank the reviewer for this question. We address it from two perspectives.
>
> **Parsing errors are rare.** Attribute-Level Data Decomposition is a lightweight structured verbatim extraction task that identifies exact substrings from captions and assigns corresponding attribute types. Manual verification on 3,000 samples yields 100% accuracy with GPT-OSS-120B (MXFP4, fully open-source, deployable on a single 80GB GPU). We additionally tested GPT-OSS-20B on 600 samples, also achieving 100% accuracy, confirming this task is well within the capability of modern LLMs.
>
> **The architecture is inherently robust to potential noise.** Even if occasional errors occur, their impact is mitigated by: (i) $L_{attr}$  works jointly with $L_{query}$ , where the holistic branch provides correct sentence-level supervision regardless of attribute-level noise; (ii) each expression yields multiple attributes, so a single error is mitigated by correct supervision from the remaining ones.
>
> ---

---

### Decision · Program_Chairs · 2026-04-30

**Decision:**

Accept (regular)

**Comment:**

All reviewers agree that the paper tackles an important problem, “enabling simultaneous multi-target detection and rich semantic grounding, which previously required separate or specialized models” [AjpB]. There were concerns about the experiments not validating the results, missing comparisons to geospatial VLMs, and explanations for efficiency gains. All of these were thoroughly covered in the extensive rebuttals, and mostly acknowledged by the reviewers. One negative reviewer, VRGf, provided a relatively terse reviewer, did not acknowledge the rebuttal, and did not provide a final justification. The other negative reviewer, 4LoK, acknowledged that their concerns were partially addressed by the extensive rebuttal but did not change his score. The AC generally agrees with the authors’ rebuttal points, leading to the paper’s acceptance.